# TokenSqueeze: Performance-Preserving Compression for Reasoning LLMs

**Yuxiang Zhang[1]\*, Zhengxu Yu[3]\*, Weihang Pan[2], Zhongming Jin[3]**
**Qiang Fu[4], Deng Cai[1], Binbin Lin[2]†, Jieping Ye[3]†**

[1]State Key Lab of CAD&CG, Zhejiang University
[2]School of Software Technology, Zhejiang University
[3]Alibaba Cloud  [4]Zhiyuan Research Institute

## Abstract

Emerging reasoning LLMs such as OpenAI-o1 and DeepSeek-R1 have achieved strong performance on complex reasoning tasks by generating long chain-of-thought (CoT) traces. However, these long CoTs result in increased token usage, leading to higher inference latency and memory consumption. As a result, balancing accuracy and reasoning efficiency has become essential for deploying reasoning LLMs in practical applications. Existing long-to-short (Long2Short) methods aim to reduce inference length but often sacrifice accuracy, revealing a need for an approach that maintains performance while lowering token costs. To address this efficiency-accuracy tradeoff, we propose TokenSqueeze, a novel Long2Short method that condenses reasoning paths while preserving performance and relying exclusively on self-generated data. First, to prevent performance degradation caused by excessive compression of reasoning depth, we propose to select self-generated samples whose reasoning depth is adaptively matched to the complexity of the problem. To further optimize the linguistic expression without altering the underlying reasoning paths, we introduce a distribution-aligned linguistic refinement method that enhances the clarity and conciseness of the reasoning path while preserving its logical integrity. Comprehensive experimental results demonstrated the effectiveness of TokenSqueeze in reducing token usage while maintaining accuracy. Notably, DeepSeek-R1-Distill-Qwen-7B fine-tuned by using our proposed method achieved a 50% average token reduction while preserving accuracy on the MATH500 benchmark. TokenSqueeze exclusively utilizes the model's self-generated data, enabling efficient and high-fidelity reasoning without relying on manually curated short-answer datasets across diverse applications. Our code is available at `https://github.com/zhangyx1122/TokenSqueeze`.

## 1 Introduction

Large Language Models (LLMs) have revolutionized artificial intelligence with their exceptional performance in complex reasoning tasks, such as mathematical problem-solving and algorithmic programming [3, 40, 18, 15, 10, 38, 39]. Recent advancements in this field primarily stem from enhancing models' ability to use extended chain-of-thought (CoT) reasoning in reinforcement learning-based self-play training [31], as seen in OpenAI-o1 [14] and DeepSeek-R1 [11]. These models leverage multi-step reasoning to simulate human cognitive strategies, including hypothesis generation, iterative refinement, and self-correction [9].

---

\*Equal contribution.
†Corresponding author.

39th Conference on Neural Information Processing Systems (NeurIPS 2025).

Although long CoT enables deep reasoning abilities, it introduces issues like increased inference latency and memory usage, making models impractical for time-sensitive or resource-constrained applications [20, 8]. Meanwhile, this has also led to an "overthinking" phenomenon, where models produce redundant reasoning steps that do not contribute meaningful value in simpler tasks, thereby impeding their effectiveness in real-world applications [36, 17, 28, 7, 34]. This is particularly evident in LLM-based agent systems, where the multi-turn trial-and-error nature necessitates swift and concise reasoning for effective interactions and timely decision-making.

Some existing methods aim to improve efficiency through inference-time compression [5, 35], using prompt-based approaches or modified decoding strategies to shorten outputs. However, these techniques often face performance limitations, as the underlying model remains unchanged. In contrast, most current train-time strategies [30, 29] promote shorter responses by incorporating penalty terms into either the reward or objective functions during training. Although effective in reducing output length, these approaches often compromise essential reasoning steps, leading to significant accuracy declines—a phenomenon known as the reasoning oversimplification dilemma.

In contrast, we argue that the concise and efficient short answers we pursue are a matter of expression preference. Our experimental results demonstrate that when the reasoning length exceeds a certain token threshold, the correlation between token number and model performance significantly weakens. Hence, the Long2Short problem can be framed as a preference learning task that teaches the model to answer in a succinct tone while preserving its reasoning depth. Furthermore, we argue that maintaining adaptive reasoning depth, tailored to each problem's complexity, is key to preserving model performance.

In this study, we investigate the Long2Short problem from two perspectives: the creation of effective long and short response pairs, and the formulation of preference learning objectives. To address the Long2Short problem, we introduce TokenSqueeze, a novel training-time preference learning method that enhances reasoning efficiency without the need for external teacher models or additional annotations. Instead, our approach leverages self-generated reasoning data to meticulously construct long and short preference pairs, ensuring scalability and resource efficiency. To generate differentiated preference pairs, we introduce a data construction methodology that employs two key techniques: (1) adaptive reasoning depth selection, which modulates reasoning depth based on problem complexity to ensure essential steps are retained; and (2) intra-step refinement, which rewrites individual reasoning steps to enhance information density while preserving their meaning. Additionally, to optimize these compact and coherent reasoning traces, we incorporate length-aware signals into a preference-based training objective, thereby promoting responses that are both concise and logically sound.

Our approach strikes a balance between efficiency and accuracy, showing that high-quality reasoning can be achieved without relying on handcrafted short-answer datasets. The key contributions of this work are as follows:

- We propose a method to generate high-quality long-short reasoning pairs by combining adaptive depth selection with intra-step linguistic refinement, without relying on external models or annotations.
- We introduce a length-aware preference objective that explicitly reinforces the model's preference for concise reasoning.
- Extensive experiments show that our method significantly improves reasoning efficiency while maintaining model performance, demonstrating its effectiveness, and broad applicability across reasoning tasks.

## 2    Related Work

### 2.1    Online Reinforcement Learning with Length Penalty

Recent advances in reasoning efficiency have explored online reinforcement learning methods like Proximal Policy Optimization (PPO) and Group Relative Policy Optimization (GRPO) [25, 26, 37]. These approaches modify the reward function in RL training to reward concise and correct responses while penalizing verbose or incorrect ones, promoting reasoning compression. Notable implementations include Kimi-k1.5 (RL) [29], which integrates a length penalty into its framework, L1 [2], which leverages reinforcement learning to satisfy prompt-specified length constraints during

training, and O1-Pruner [21], which introduces a length-harmonizing reward mechanism based on the length ratio between reference and predicted chain-of-thought sequences.

While effective, online RL methods are computationally expensive due to the need for response sampling at each training step. This increases overhead and limits their scalability for large model training. Moreover, large language models require many samples to generate efficient reasoning traces, further raising the computational burden.

## 2.2 Offline Data-Driven Optimization Algorithms

In contrast to online methods, offline optimization approaches rely on existing response datasets, avoiding repeated sampling during training and offering substantial computational savings. They typically follow a two-stage process: first constructing concise reasoning datasets, then training the model via supervised fine-tuning or direct preference optimization [24].

Several studies have focused on constructing training datasets by identifying the shortest correct responses from base model outputs. For instance, Kimi-k1.5 (DPO) [29] and Sky-T1-Flash [30] construct training sets by selecting minimal-length correct samples from response pools, while Self-Training [22] employs few-shot prompting to guide models toward generating shorter answers before selection. Token-Budget [12] adopts a prompt-search strategy to generate maximally concise reasoning samples, and DAST [27] implements length-aware reward shaping to select preferred training samples. These methods reduce token usage by selecting the shortest correct responses, but this often leads to overly aggressive compression that removes essential reasoning steps and harms accuracy. In addition, relying solely on unaltered model outputs may miss more concise yet high-quality traces due to sampling variability. TokenSqueeze addresses both issues by combining adaptive depth selection with trace refinement to improve training quality.

Alternative methods improve the quality of training data by rephrasing model-generated responses, which are then used as supervision in the training process. These methods avoid the need for extensive sampling through two primary strategies: prompt-based rephrasing with external LLMs, as demonstrated by C3oT [16], which employs GPT-4 [1] for response refinement, and rule-based content pruning exemplified by TokenSkip [33], which uses importance-based token elimination, along with Learn-to-Skip [19], which integrates step merging and skipping mechanisms. While efficient, these refinement methods can inadvertently remove important context or break coherence, hurting reasoning quality. TokenSqueeze addresses this trade-off by rewriting reasoning steps into denser forms under KL constraints, preserving information without external models.

# 3 Methodology

TokenSqueeze optimizes reasoning efficiency with a three-stage training method that balances logical accuracy and linguistic conciseness. First, we adaptively filter self-generated reasoning traces to maintain the appropriate depth (Section 3.1). Next, we refine individual reasoning steps to compress them while preserving meaning (Section 3.2). Finally, we optimize the model using a composite objective that promotes both correctness and brevity (Section 3.3).

## 3.1 Adaptive Reasoning Depth Selection

Our data construction pipeline starts by generating diverse reasoning traces through self-sampling from the base language model. Unlike existing methods that simply select the shortest correct responses [29, 30, 6, 27], we introduce an adaptive reasoning depth selection method that ensures the appropriate trace length. Maintaining an appropriate reasoning depth is crucial for achieving high accuracy, especially on complex problems. Previous studies have shown that while there is an optimal range of chain lengths for high accuracy, this range shifts toward longer chains as problems become more difficult [32].

We formalize the selection process using a dynamic quantile mechanism. Let $\alpha$ denote a tunable hyperparameter, and define $p = \frac{c}{N}$ as the fraction of correct responses among $N$ total responses, with $c$ representing the number of correct samples. The adaptive quantile is computed as $q = \alpha \cdot (1 - p)$, enabling the preferred chain length to vary naturally with problem difficulty.

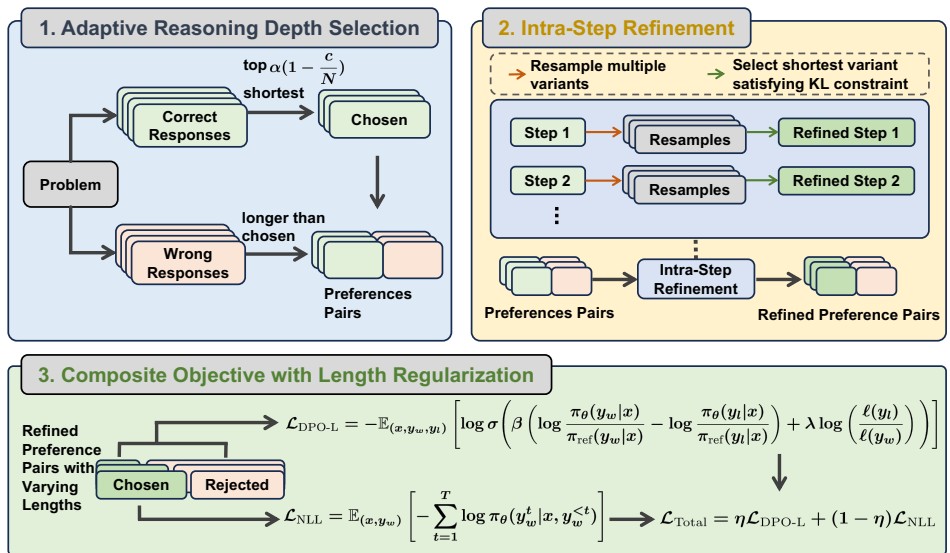

Figure 1: Overview of the TokenSqueeze. It first selects self-generated responses based on adaptive reasoning depth, then rewrites them under KL constraints to improve information density, and finally trains the model with a composite objective that promotes accuracy and brevity.

For each problem, we first sort all correct reasoning traces by token length, forming an ordered set $S = \{\tau_1, \tau_2, \ldots, \tau_c\}$ where $\tau_1$ is the shortest and $\tau_c$ the longest. The selection threshold index is determined by $k = \lceil q \cdot c \rceil$, and the subset $\{\tau_1, \ldots, \tau_k\}$ is chosen as the set of preferred positive examples. Each selected trace $\tau_i$ is further paired with longer incorrect responses to construct contrastive preference samples, with up to $M$ preference pairs generated per problem to maintain data diversity, where we set $M = 64$ in our experiments.

This adaptive selection has two key advantages: (1) For easier problems with higher correctness rates, it favors shorter reasoning chains, and (2) for harder problems with lower correctness rates, it retains longer chains to capture critical logical steps. Overall, the resulting dataset ensures balanced coverage across different difficulty levels while optimizing reasoning efficiency.

### 3.2 Intra-Step Linguistic Refinement via Distributional Alignment

Building on the preference pair construction from Section 3.1, we further improve the conciseness of training samples through a systematic intra-step linguistic refinement method. While existing methods typically use rule-based truncation or LLM-based rewriting to compress reasoning traces, they often fail to guarantee logical integrity and can introduce information loss. To address these issues, we propose an intra-step linguistic refinement method that minimizes information loss while effectively reducing length.

Given a generated reasoning trace $\mathcal{A} = (p, s_1, \ldots, s_N)$, consisting of a prompt $p$ and a sequence of reasoning steps $\{s_i\}_{i=1}^N$, we aim to compress each step while preserving its informational content. For each reasoning step $s_i$, we first resample a set of $K$ candidate rewrites $\{s_i^{(k)}\}_{k=1}^K$, conditioned on the preceding context $\mathcal{A}_{<i} = (p, s_1, \ldots, s_{i-1})$, where we set $K = 64$. Among these candidates, we select the one that minimizes token length while satisfying a KL divergence constraint to preserve downstream semantics. The optimization for each step can be formulated as:

$$\min_{s_i' \in \{s_i^{(k)}\}} \ell(s_i') \quad \text{subject to} \quad D_{\mathrm{KL}}\left(P_\theta(\cdot|p, s_{\leq i}) \,\|\, P_\theta(\cdot|p, s_{<i}, s_i')\right) < \varepsilon, \tag{1}$$

where $\ell(s_i')$ represents the token length of the candidate, and $\varepsilon > 0$ is a tunable threshold that controls the level of information preservation. Here, $P_\theta$ denotes the probability distribution over all possible future responses, conditioned on the model parameters $\theta$.

**Question**

In a press conference before the All-Star Game, eight All-Stars will be taking questions. Three are Cubs, three are Red Sox, and two are Yankees. If teammates insist on sitting together, how many ways can the eight All-Stars be seated in a row?

**Original COT**

To determine the number of ways the eight All-Stars can be seated in a row with each team sitting together, we can treat each team as a single unit or "block."
First, we calculate the number of ways to arrange the three blocks (Cubs, Red Sox, Yankees). This is a permutation of three distinct blocks, which is 3! = 6.
Next, within each block, the team members can be arranged among themselves.
- The Cubs have 3 members, so they can be arranged in 3! = 6 ways.
- The Red Sox also have 3 members, so they can be arranged in 3! = 6 ways.
- The Yankees have 2 members, so they can be arranged in 2! = 2 ways.
To find the total number of arrangements, we multiply the number of ways to arrange the blocks by the number of ways to arrange the members within each block: 6 (for the blocks) × 6 (Cubs) × 6 (Red Sox) × 2 (Yankees) = 432.

**Rewrite COT**

To determine the number of ways the eight All-Stars can be seated in a row with each team sitting together, we can treat each team as a single unit or "block."
First, we have three blocks: one for each team (Cubs, Red Sox, Yankees). The number of ways to arrange these three blocks is 3! = 6.
Next, within each block, the team members can be arranged among themselves. The Cubs can be arranged in 3! ways, the Red Sox in 3! ways, and the Yankees in 2! ways.
Multiplying these together gives the total number of arrangements: 3! × 3! × 3! × 2! = 6 × 6 × 6 × 2 = 432.

Figure 2: Example of reasoning trace before and after intra-step linguistic refinement. The refined version achieves higher information density while preserving logical integrity.

To estimate the KL divergence across full distributions, we apply a local token window approximation, with the detailed derivation provided in the Appendix:

$$D_{\mathrm{KL}}^{\mathrm{full}} \approx \sum_{j=1}^{\min(T,L)} D_{\mathrm{KL}}\left(Q_\theta(\cdot \mid p, s_{\leq i}, t_{1:j-1}) \,\|\, Q_\theta(\cdot \mid p, s_{<i}, s_i', t_{1:j-1})\right). \tag{2}$$

Here, $D_{\mathrm{KL}}^{\mathrm{full}}$ corresponds to the KL divergence term defined in Equation 1. The index $t_j$ denotes the $j$-th token in the consecutive reasoning steps following the $i$-th step $s_i$. In particular, $t_1$ corresponds to the first token of step $s_{i+1}$. If the tokens within step $s_{i+1}$ are insufficient to fill the token window, the sequence continues by accumulating tokens from subsequent reasoning steps until the window reaches its length limit or the response ends. $Q_\theta$ represents the conditional probability distribution over the vocabulary at position $j$, given all preceding tokens. $T$ denotes the total number of tokens in the following reasoning segment, and $L = 512$ is the fixed window size used to maintain computational tractability.

More examples of refined reasoning traces are provided in the appendix for reference. Our refinement method offers three key advantages over previous methods: (1) it ensures information preservation through explicit KL divergence constraints, (2) it enables adaptive compression by adjusting the ratio based on local context importance, and (3) it is model-agnostic, avoiding reliance on external LLMs. Together, these features allow TokenSqueeze to achieve efficient and accurate reasoning trace compression. Illustrative examples of refined reasoning traces, demonstrating improved clarity and information density, are provided in the appendix.

### 3.3 Composite Optimization Objective

The final component of TokenSqueeze jointly optimizes the language model for reasoning fidelity and conciseness through a composite preference-based objective. Building upon standard Direct Preference Optimization (DPO), we introduce an adaptive length-aware margin to encourage efficient reasoning traces without sacrificing logical correctness.

We define the underlying reward maximization as:

$$\max_{\pi_\theta} \mathbb{E}_{x,y}\left[r_\phi(x, y) - \beta D_{\mathrm{KL}}(\pi_\theta \| \pi_{\mathrm{ref}}) - \lambda \log \ell(y)\right], \tag{3}$$

where $\ell(y)$ denotes the token length of response $y$, $\lambda$ controls the strength of length regularization, and $\beta$ regulates the divergence penalty from the reference policy $\pi_{\mathrm{ref}}$.

Transforming this reward into a preference loss yields the DPO-L (Direct Preference Optimization with Length-aware) objective:

$$\mathcal{L}_{\text{DPO-L}} = -\mathbb{E}_{(x,y_w,y_l)} \left[ \log \sigma \left( \beta \left( \log \frac{\pi_\theta(y_w|x)}{\pi_{\text{ref}}(y_w|x)} - \log \frac{\pi_\theta(y_l|x)}{\pi_{\text{ref}}(y_l|x)} \right) + \lambda \log \left( \frac{\ell(y_l)}{\ell(y_w)} \right) \right) \right], \quad (4)$$

where $(x, y_w, y_l)$ denotes the input along with the preferred (shorter, correct) and rejected (longer or incorrect) responses. The additional logarithmic term $\log(\ell(y_l)/\ell(y_w))$ adaptively scales the margin based on the relative length difference, strengthening preference signals for pairs exhibiting greater compression gains while preserving standard behavior for comparable-length pairs.

To further stabilize training and prevent reward collapse on preferred responses during optimization, we incorporate supervised fine-tuning on positive examples, resulting in the final composite objective:

$$\mathcal{L}_{\text{Total}} = \eta \mathcal{L}_{\text{DPO-L}} + (1 - \eta)\mathbb{E}_{(x,y_w)} \left[ -\sum_{t=1}^{T} \log \pi_\theta(y_w^t|x, y_w^{<t}) \right], \quad (5)$$

where we set $\eta = 0.5$ to balance the two components.

This composite training formulation maintains the theoretical guarantees of DPO while explicitly promoting compact, high-fidelity reasoning traces. Combined with adaptive sample selection and intra-step refinement, it completes the TokenSqueeze method for efficient reasoning optimization.

## 4   Experiments

### 4.1   Experimental Setup

#### 4.1.1   Training Configurations

We train DeepSeek-R1-Distill-Qwen-7B and DeepSeek-R1-Distill-Qwen-1.5B [11] models using the TokenSqueeze method described in Sections 3.1, 3.2, and 3.3. All models are optimized with a learning rate of $5 \times 10^{-6}$ and a batch size of 128, using the Adam optimizer. Training is conducted using the PyTorch framework on computing nodes equipped with $8 \times$NVIDIA Tesla A100 GPUs. Additional training details are provided in the appendix.

#### 4.1.2   Datasets and Evaluation Metrics

We evaluate model performance on four benchmark datasets: AIME24, MATH500 [13], AIME25, and LiveCodeBench [23]. As comparisons, we include several recent strong methods such as DAST [27], TrainEffi [4]. Notably, we also compared with a reproduced version of Kimi-k1.5 (DPO) [29] following the experimental setup described in its official release, since the original implementation is not released and the reported results are not based on Qwen-2.5 7B/1.5B model. In our result tables, "baseline" refers to the unaugmented original model prior to applying any enhancements or the TokenSqueeze method.

We report all results as the average of 16 independent runs to reduce variance and ensure robustness. A sampling temperature of 0.6 is used consistently across AIME24, MATH500, and AIME25 to ensure fair comparison. For LiveCodeBench, we evaluate on problems dated from 2024.08.01 to 2025.01.31, following the setup used by DeepSeek-R1 [11]. This time window is chosen because it contains fewer examples affected by data leakage, making it a more reliable benchmark for real-world performance.

To evaluate model performance, we use four key metrics: Answer Accuracy, the percentage of correctly solved problems; Average Length of Correct Responses (Len-T), the average number of tokens in correct answers; Average Length of All Responses (Len-A), the average number of tokens across all generated responses; and Area Under the Curve (AUC), which measures performance under a 32K token budget by computing the area under the accuracy–token usage curve.

Table 1: Performance comparison of different training methods across two model sizes. TokenSqueeze consistently improves token efficiency while maintaining or improving accuracy, achieving higher AUC scores compared to baselines. Bolded metrics in the table indicate our method.

| Dataset | Method | DeepSeek-R1-Distill-Qwen-7B | | | | DeepSeek-R1-Distill-Qwen-1.5B | | | |
|---|---|---|---|---|---|---|---|---|---|
| | | Acc (%) | Len-T | Len-A | AUC (%) | Acc (%) | Len-T | Len-A | AUC (%) |
| **AIME24** | Baseline | 55.5 | 7543 | 13337 | 41.6 | 28.9 | 7374 | 16906 | 22.5 |
| | Kimi-k1.5 (reproduced) [29] | 51.2 | 5249 (-30.4%) | 9221 (-30.9%) | 41.8 | 28.1 | 5034 (-31.7%) | 12159 (-28.1%) | 23.8 |
| | DAST [27] | 53.3 | 6339 (-16.0%) | – | – | – | – | – | – |
| | TrainEffi [4] | 56.0 | – | 10768 (-19.2%) | – | 31.7 | – | 9399 (-44.4%) | – |
| | **TokenSqueeze** | **57.5** | **5157 (-31.6%)** | **9189 (-31.1%)** | **48.5** | **33.3** | **5841 (-20.8%)** | **10731 (-36.5%)** | **27.4** |
| **MATH500** | Baseline | 92.8 | 3638 | 4190 | 83.6 | 83.9 | 3637 | 5412 | 76.1 |
| | Kimi-k1.5 (reproduced) | 88.2 | 1698 (-53.3%) | 2298 (-45.2%) | 83.7 | 80.8 | 1870 (-48.6%) | 3029 (-44.0%) | 76.2 |
| | DAST | 92.6 | 2802 (-23.0%) | – | – | – | – | – | – |
| | TrainEffi | 92.3 | – | 3259 (-22.2%) | – | 82.7 | – | 2818 (-47.9%) | – |
| | **TokenSqueeze** | **92.4** | **1773 (-51.3%)** | **2045 (-51.2%)** | **87.5** | **83.2** | **2005 (-44.9%)** | **2750 (-49.2%)** | **78.1** |
| **AIME25** | Baseline | 39.2 | 6646 | 14372 | 31.2 | 24.4 | 5818 | 16070 | 20.1 |
| | Kimi-k1.5 (reproduced) | 38.1 | 4337 (-34.7%) | 10575 (-26.4%) | 33.1 | 23.3 | 4528 (-22.2%) | 13991 (-12.9%) | 20.1 |
| | DAST[1] | 39.1 | 4602 (-30.8%) | – | – | – | – | – | – |
| | **TokenSqueeze** | **39.8** | **4711 (-29.1%)** | **10550 (-26.6%)** | **34.1** | **24.4** | **4672 (-19.7%)** | **11169 (-30.5%)** | **20.9** |
| **LiveCode Bench** | Baseline | 31.3 | 3961 | 20690 | 27.5 | 13.4 | 2629 | 26513 | 12.3 |
| | Kimi-k1.5 (reproduced) | 24.8 | 3256 (-17.8%) | 19242 (-7.0%) | 22.3 | 12.5 | 2088 (-20.6%) | 25318 (-4.5%) | 12.6 |
| | DAST[1] | 29.7 | 3494 (-11.8%) | – | – | – | – | – | – |
| | **TokenSqueeze** | **35.0** | **3200 (-19.2%)** | **15635 (-24.4%)** | **31.6** | **16.7** | **2587 (-1.6%)** | **24842 (-6.3%)** | **15.4** |

[1] Reproduced from the official open-source model, as the original paper did not report this result.

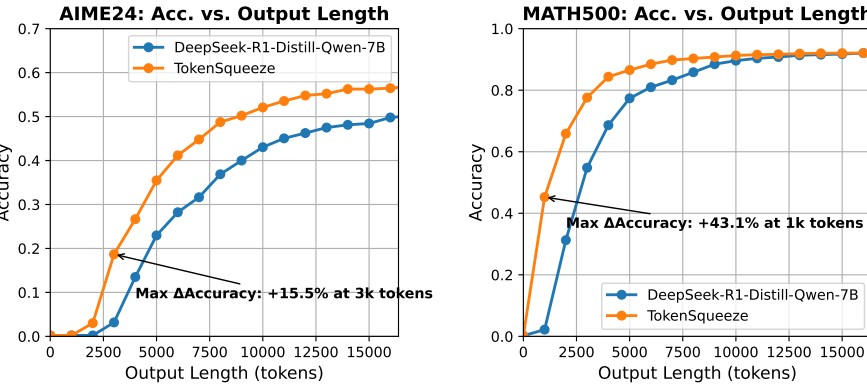

Figure 3: TokenSqueeze outperforms the base model across token budgets on AIME24 and MATH500, with up to 15.5% higher accuracy on AIME24 (3K tokens) and 43.1% on MATH500 (1K tokens).

## 4.2 Evaluation on General Reasoning Benchmarks

Table 1 summarizes the performance of TokenSqueeze compared to both base models and recent strong methods across four benchmark datasets and two model scales. Overall, TokenSqueeze consistently improves accuracy while significantly reducing token consumption, leading to better token efficiency and higher AUC scores.

On the mathematical reasoning datasets AIME24, MATH500, and AIME25, TokenSqueeze compresses token usage significantly without sacrificing accuracy. In MATH500, it reduces the average length of correct reasoning traces by nearly half, while maintaining accuracy close to the base model. Similarly, on AIME24 and AIME25, it not only compresses the output effectively but also improves or matches the base model's accuracy, outperforming methods like Kimi-k1.5 (DPO) [29] and DAST [27] in both efficiency and effectiveness. Notably, TokenSqueeze demonstrates stronger compression performance on the 7B model compared to the 1.5B model across most metrics.

Beyond mathematical reasoning, TokenSqueeze also excels in programming tasks, demonstrating its ability to generalize. On LiveCodeBench, which includes real-world coding challenges, Token-

Squeeze significantly improves accuracy and reduces output length, showing its robustness and adaptability across domains.

In terms of AUC, which measures accuracy under token budget constraints, TokenSqueeze consistently improves performance across all datasets, proving it to be an effective solution for enhancing reasoning quality while reducing computational costs. Figure 3 further illustrates the efficiency gains: under the same token budget, TokenSqueeze achieves up to 15.5% higher accuracy on AIME24 at 3k tokens and 43.1% higher accuracy on MATH500 at 1k tokens compared to the base model. These results highlight TokenSqueeze's success in balancing reasoning quality and efficiency, addressing the long-standing trade-off between accuracy and computational cost in LLM tasks.

## 4.3 Ablation Study

### 4.3.1 Impact of Adaptive Reasoning Depth Selection

We first evaluate the contribution of TokenSqueeze's adaptive selection mechanism, which adjusts the threshold for selecting appropriate reasoning depths based on problem complexity. As shown in Table 2, we compare four configurations for selecting training pairs from self-generated responses: Shortest, Q-FIX, Q-DYN (w/ extra pos), and Q-DYN.

All configurations use the same preference-based learning method, where a correct response is selected as the positive sample and paired with longer incorrect responses. In the Shortest baseline, only the shortest correct trace is selected. Q-FIX selects correct responses at a fixed quantile of the length ranking, without adapting to problem difficulty. Q-DYN introduces an adaptive threshold, $q = \alpha \times (1 - p)$, where $p$ is the correctness rate and $\alpha = 0.2$, allowing the reasoning depth to vary with task complexity. Q-DYN (w/ extra pos) extends Q-DYN by including all correct traces longer than 1.5× the positive sample length—as in the Kimi-k1.5 (DPO) [29] setting—as additional negative samples.

Compared to Shortest, Q-FIX achieves higher accuracy with only a moderate increase in response length, suggesting that avoiding excessive compression is beneficial. Q-DYN (w/ extra pos) further reduces response length, but at the cost of accuracy. We believe this accuracy drop is due to the inclusion of correct traces as negative samples, which weakens the model's ability to distinguish correct from incorrect reasoning and encourages excessive compression during training. In contrast, Q-DYN achieves the highest accuracy on both datasets while maintaining competitive length, supporting our hypothesis that adapting reasoning depth to task complexity is crucial for balancing efficiency and correctness [32].

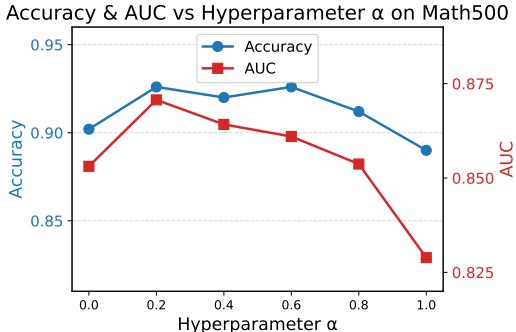

To further explore the impact of the $\alpha$ parameter, we conduct a comprehensive analysis and visualize the results in Figure 4. We find that $\alpha = 0.2$ strikes the best balance between accuracy and token efficiency across both datasets. Extreme values, such as $\alpha = 0$ or $\alpha = 1$, degrade performance due to either excessive compression or overly long reasoning chains that introduce noise and irrelevant logic.

Figure 4: Effect of the $\alpha$ parameter in adaptive quantile selection. Moderate values (e.g., $\alpha = 0.2$) provide the best balance between accuracy and token efficiency, while extreme values degrade performance.

### 4.3.2 Impact of Intra-Step Linguistic Refinement

Building on the preference data from Section 3.1, we now evaluate the effectiveness of intra-step linguistic refinement in improving reasoning efficiency. We compare four rewriting methods: (1) TokenSqueeze without refinement, (2) prompt-based rewriting using GPT-4o-mini, (3) the TokenSkip method from prior work, and (4) our proposed refinement method. As shown in Table 3, our method achieves higher accuracy on both AIME24 and MATH500, performing similarly to the No Refinement baseline. Notably, both GPT-4o-mini and TokenSkip are less effective at reducing response length,

Table 2: Ablation results comparing different preference data configurations. Among all variants, Q-DYN, the method adopted by TokenSqueeze, achieves the best overall performance.

| Method | AIME24 | | | MATH500 | | |
|--------|--------|-----|---------|---------|-----|---------|
| | Acc (%) | Len | AUC (%) | Acc (%) | Len | AUC (%) |
| Shortest | 53.3 | 5960 | 43.7 | 90.8 | 1926 | 85.5 |
| Q-FIX | 55.0 | 6126 | 44.8 | 92.2 | 2054 | 86.5 |
| Q-DYN (w/ extra pos) | 52.3 | 5666 | 43.3 | 90.8 | 1742 | 86.0 |
| **Q-DYN** | **57.3** | **6190** | **46.5** | **92.8** | **2180** | **86.7** |

Table 3: TokenSqueeze achieves the highest accuracy by reducing both the number and length of reasoning steps, demonstrating its effectiveness under compression and outperforming other methods.

| Method | AIME24 | | | | | MATH500 | | | | |
|--------|--------|-----|---------|-------|---------|---------|-----|---------|-------|---------|
| | Acc (%) | Len | AUC (%) | Steps | StepLen | Acc (%) | Len | AUC (%) | Steps | StepLen |
| Baseline | 55.5 | 7543 | 41.6 | 267.0 | 28.1 | 92.8 | 3638 | 83.6 | 100.2 | 34.8 |
| No Refinement | 57.3 | 6190 | 46.5 | 198.5 | 29.1 | 92.8 | 2180 | 86.7 | 55.0 | 35.1 |
| 4o-mini Rewrite | 39.1 | 6596 | 31.3 | – | – | 79.0 | 3333 | 56.4 | – | – |
| TokenSkip Rewrite | 54.4 | 6378 | 43.8 | – | – | 84.3 | 2686 | 64.1 | – | – |
| **TokenSqueeze** | **57.5** | **5157** | **48.5** | **194.4** | **26.3** | **92.4** | **1773** | **87.5** | **57.5** | **30.6** |

likely due to diminished reasoning ability. This confirms that our refinement method preserves essential information while shortening responses, offering a better balance between conciseness and correctness.

To better understand how compression is achieved, we break down the total response length into two components: the number of reasoning steps and the average tokens per step. The method without refinement significantly reduces the number of steps but slightly increases the average length per step, suggesting that compression primarily occurs along the reasoning depth axis. In contrast, our full method achieves further reduction by shortening individual steps, highlighting the effectiveness of linguistic-level refinement.

### 4.3.3 Impact of Composite Optimization Objectives

We further evaluate the effectiveness of TokenSqueeze's multi-objective training strategy by comparing four optimization variants, all trained on the same dataset: (1) Direct Preference Optimization (DPO), (2) supervised fine-tuning (SFT) on selected samples only, (3) training with a combined DPO and SFT loss, and (4) our full method, combining SFT loss with length-regularized DPO loss.

As shown in Table 4, our analysis reveals that pure DPO training, while effective at reducing reasoning length, significantly degrades reasoning accuracy. Without safeguards, the reward for preferred samples diminishes over time, leading to unstable, low-quality outputs. In contrast, SFT training on positive samples maintains reasoning accuracy but achieves only modest reductions in token usage, as it lacks mechanisms to penalize verbosity.

Combining DPO loss and SFT loss partially addresses these issues, recovering much of the accuracy loss while enabling more effective compression. In this setup, the SFT loss helps stabilize training, while the DPO loss encourages concise outputs. However, further analysis suggests that there is still room for improvement, particularly in terms of better leveraging the dataset's compression potential. Our complete approach, which integrates SFT loss with a length-regularized DPO loss, delivers the strongest performance by balancing accuracy with controlled output length. By explicitly incorporating length signals into the optimization objective, TokenSqueeze effectively fine-tunes the trade-off between brevity and logical completeness.

These findings underscore the importance of TokenSqueeze's integrated training strategy. The SFT loss provides stability during training, the DPO loss optimizes the model using preference signals, and the length regularization component reinforces the model's tendency to generate more concise responses.

Table 4: Comparison of training objectives using the same data. Our method combining SFT and length-regularized DPO achieves the best balance of accuracy and compression.

| Method | AIME24 | | | MATH500 | | |
|---|---|---|---|---|---|---|
| | Acc (%) | Len | AUC (%) | Acc (%) | Len | AUC (%) |
| DPO | 48.3 | 4300 | 42.0 | 91.6 | 1974 | 86.1 |
| SFT | 56.0 | 5734 | 46.3 | 91.8 | 2271 | 85.5 |
| DPO+SFT | 57.0 | 5420 | 47.3 | 92.6 | 1865 | 87.4 |
| **TokenSqueeze** | **57.5** | **5157** | **48.5** | **92.4** | **1773** | **87.5** |

## 5 Limitation

While TokenSqueeze achieves substantial efficiency gains without compromising reasoning accuracy, several limitations remain that we aim to address in future work.

**Heuristic Hyperparameter Selection.** Some hyperparameters—most notably the KL threshold $\varepsilon$ used during intra-step rewriting—were determined heuristically based on limited preliminary experiments rather than systematic tuning. Although we conducted multiple experiments with varying hyperparameter configurations to thoroughly validate the effectiveness of our method, we did not have sufficient time or computational resources to perform exhaustive hyperparameter optimization. We have not yet explored the sensitivity or mutual interactions of these parameters in depth. In particular, $\varepsilon$ governs the trade-off between semantic fidelity and linguistic brevity: setting it too low can restrict compression excessively, while setting it too high may cause semantic drift. A more principled study of this balance, combined with adaptive or data-driven tuning strategies, could improve robustness across tasks and model scales. In future work, we plan to develop an adaptive mechanism that automatically adjusts $\varepsilon$ based on context difficulty and local token divergence.

**Offline-only Setting.** Our current framework operates entirely under an offline preference optimization paradigm, where preference pairs are pre-generated and the model is trained without real-time interaction with the environment. This design simplifies implementation and ensures training stability, but it limits adaptivity—the model cannot continuously refine its reasoning strategy based on new feedback or task distribution shifts. In the future, we plan to extend TokenSqueeze into an online reinforcement learning setting, where preference generation, policy updates, and reward estimation are jointly optimized. Such an extension could enable continual self-improvement and stronger alignment between reasoning efficiency and task accuracy.

## 6 Conclusion

In this paper, we introduce TokenSqueeze, a method for compressing reasoning traces in large language models using only self-generated training data, without manual annotations or teacher models. TokenSqueeze addresses the reasoning oversimplification dilemma by combining adaptive reasoning depth selection, intra-step linguistic refinement, and length-aware preference optimization to maintain task performance. Extensive experiments on reasoning benchmarks, including math and code tasks, show that TokenSqueeze significantly reduces token usage while preserving or even improving accuracy, offering a practical path toward efficient, high-fidelity reasoning for real-world deployment.

## Acknowledgements

This work was supported in part by the Key R&D Program of Zhejiang Province (No. 2025C01212) and in part by the Yongjiang Talent Introduction Programme (No. 2022A-240-G).

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

# A  Hyperparameter Settings and Implementation Details

## A.1  Data Construction and Compression Settings

We generate self-sampled reasoning traces from the base model using a sampling temperature of 0.9 to promote output diversity. During the intra-step linguistic refinement stage, each reasoning step is resampled at a temperature of 1.0 to produce 64 candidate rewrites. From these, the shortest variant is selected, provided it satisfies a KL divergence constraint of 0.005, which we determined through empirical tuning. This ensures that downstream semantics are preserved while improving overall conciseness.

Throughout data collection and training, we adopt a consistent prompting format that explicitly separates the model's reasoning trace from its final answer. The prompt template is shown below:

```
A conversation between User and Assistant. The user asks a question, and the
    Assistant solves it. The assistant first thinks about the reasoning process in
    the mind and then provides the user with the answer. The reasoning process and
    answer are enclosed within <think> </think> and <answer> </answer> tags,
    respectively, i.e., <think> reasoning process here </think> <answer> **Final
    Answer:**\n\boxed{{}} </answer>
<|User|>{question}.
<|Assistant|>
```

To construct preference pairs, refined traces are selected using our adaptive depth selection strategy. The selection threshold is dynamically determined by the formula $q = \alpha(1 - p)$, where $p$ is the correctness rate and $\alpha$ is set to 0.2. This ensures that reasoning depth adapts to problem difficulty, balancing informativeness and brevity.

## A.2  Training Configurations

We fine-tune both DeepSeek-R1-Distill-Qwen-7B and 1.5B using full-parameter training. Our implementation builds on the DPO pipeline from the LLaMAFactory framework, with the following modifications:

- Addition of a combined training objective using both supervised fine-tuning (SFT) loss and DPO loss, weighted equally at 0.5 each;

- Integration of a length-aware DPO objective (DPO-L) with the length penalty coefficient $\lambda = 1$, to explicitly promote concise outputs.

We set the learning rate to $5 \times 10^{-6}$, use a batch size of 128, and configure the maximum context length to 9000 tokens to fully capture long prompt–response pairs. Training is conducted on $8 \times$ NVIDIA A100 GPUs and completes within approximately one day.

## A.3  Evaluation Protocols

We evaluate our models on AIME24, MATH500, AIME25, and LiveCodeBench. For AIME24, MATH500, and AIME25, we use a decoding temperature of 0.6 and average results over 16 independent runs. In all cases, the maximum token generation limit is set to 32,768.

For LiveCodeBench, we follow the evaluation setup from DeepSeek-R1 and restrict evaluation to problems released between August 1, 2024 and January 31, 2025 to minimize data leakage. We use a decoding temperature of 0.2 and again average over 16 runs to ensure robustness.

# B  Proof from Equation (1) to Equation (2)

**Notation and Setup.**    We restate the two KL divergence expressions used in the main text:

$$D_{\mathrm{KL}}\big(P_\theta(\cdot \mid p, s_{\leq i}) \,\big\|\, P_\theta(\cdot \mid p, s_{<i}, s'_i)\big) \tag{1}$$

$$D_{\text{KL}}^{\text{full}} \approx \sum_{j=1}^{\min(T,L)} D_{\text{KL}}\big(Q_\theta(\cdot \mid p, s_{\leq i}, t_{1:j-1}) \,\big\|\, Q_\theta(\cdot \mid p, s_{<i}, s_i', t_{1:j-1})\big). \tag{2}$$

These quantities measure how the model's continuation distribution changes after rewriting a reasoning step $s_i$.

We use the following notation:

- $p$: input prompt.
- $s_i$: the $i$-th reasoning step.
- $s_{\leq i}$: the reasoning trace up to and including step $i$.
- $s_i'$: the rewritten version of step $s_i$.
- $t_j$: the $j$-th token in the continuation sequence after $s_i$ or $s_i'$.
- $t_{1:T}$: a continuation of length $T$ tokens.

The probability distributions are:

$$Q_\theta(\cdot \mid \cdot) : \text{token-level next-token distribution,}$$
$$P_\theta(\cdot \mid \cdot) : \text{sequence-level distribution over } t_{1:T}.$$

For brevity, define:

$$\mathsf{A}(t_{1:T}) := P_\theta(t_{1:T} \mid p, s_{\leq i}), \quad \mathsf{B}(t_{1:T}) := P_\theta(t_{1:T} \mid p, s_{<i}, s_i').$$

**Step 1: Sequence-Level KL Expansion.** By definition,

$$D_{\text{KL}}(\mathsf{A}\|\mathsf{B}) = \mathbb{E}_{t_{1:T} \sim \mathsf{A}}\left[\log \frac{\mathsf{A}(t_{1:T})}{\mathsf{B}(t_{1:T})}\right]. \tag{6}$$

Expanding both $\mathsf{A}$ and $\mathsf{B}$ autoregressively using $Q_\theta$:

$$\mathsf{A}(t_{1:T}) = \prod_{j=1}^{T} Q_\theta(t_j \mid p, s_{\leq i}, t_{1:j-1}), \quad \mathsf{B}(t_{1:T}) = \prod_{j=1}^{T} Q_\theta(t_j \mid p, s_{<i}, s_i', t_{1:j-1}).$$

Substituting yields:

$$D_{\text{KL}}(\mathsf{A}\|\mathsf{B}) = \mathbb{E}_{t_{1:T} \sim \mathsf{A}}\left[\sum_{j=1}^{T} \log \frac{Q_\theta(t_j \mid p, s_{\leq i}, t_{1:j-1})}{Q_\theta(t_j \mid p, s_{<i}, s_i', t_{1:j-1})}\right]$$

$$= \sum_{j=1}^{T} \mathbb{E}_{t_{1:j-1} \sim \mathsf{A}}\left[D_{\text{KL}}\big(Q_\theta(\cdot \mid p, s_{\leq i}, t_{1:j-1})\big\|Q_\theta(\cdot \mid p, s_{<i}, s_i', t_{1:j-1})\big)\right]. \tag{7}$$

Thus, the sequence-level KL is the expected sum of per-token conditional KL divergences.

**Step 2: Monte Carlo Approximation.** In practice, the full expectation over all token prefixes $t_{1:j-1}$ is intractable. We approximate it using a single sampled trajectory $t_{1:T} \sim \mathsf{A}$:

$$D_{\text{KL}}^{\text{full}} \approx \sum_{j=1}^{T} D_{\text{KL}}\big(Q_\theta(\cdot \mid p, s_{\leq i}, t_{1:j-1})\big\|Q_\theta(\cdot \mid p, s_{<i}, s_i', t_{1:j-1})\big). \tag{8}$$

**Step 3: Token-Window Truncation.** To further reduce computation, we truncate the summation to a fixed window size $L$ (e.g., 512 tokens):

$$D_{\mathrm{KL}}^{\mathrm{full}} \approx \sum_{j=1}^{\min(T,L)} D_{\mathrm{KL}}\big(Q_\theta(\cdot \mid p, s_{\leq i}, t_{1:j-1}) \big\| Q_\theta(\cdot \mid p, s_{<i}, s_i', t_{1:j-1})\big). \tag{9}$$

This yields Equation (2) in the main text. The approximation incurs a residual term

$$R_L = \sum_{j>L} \mathbb{E}_{t_{1:j-1} \sim \mathsf{A}}[D_{\mathrm{KL}}(\cdot \| \cdot)] \geq 0,$$

which remains small in practice since the influence of rewriting $s_i$ decays over long continuations.

## C Refined Reasoning Trace Examples

To complement our main results, we provide several illustrative examples of refined reasoning traces in this appendix. These examples demonstrate how TokenSqueeze compresses reasoning steps while preserving core logic and improving clarity.

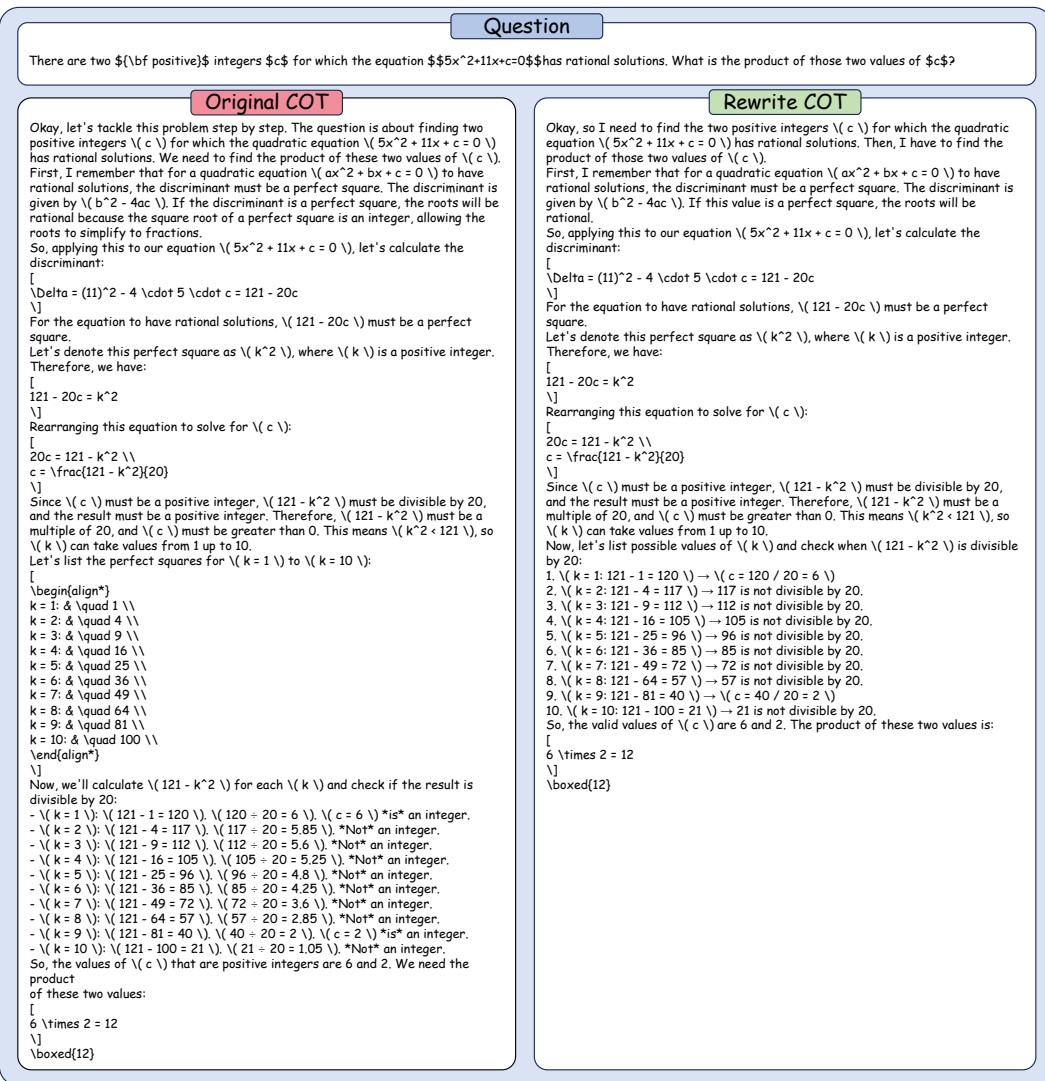

Figure 5: Reasoning trace comparison for a quadratic equation problem before and after refinement.

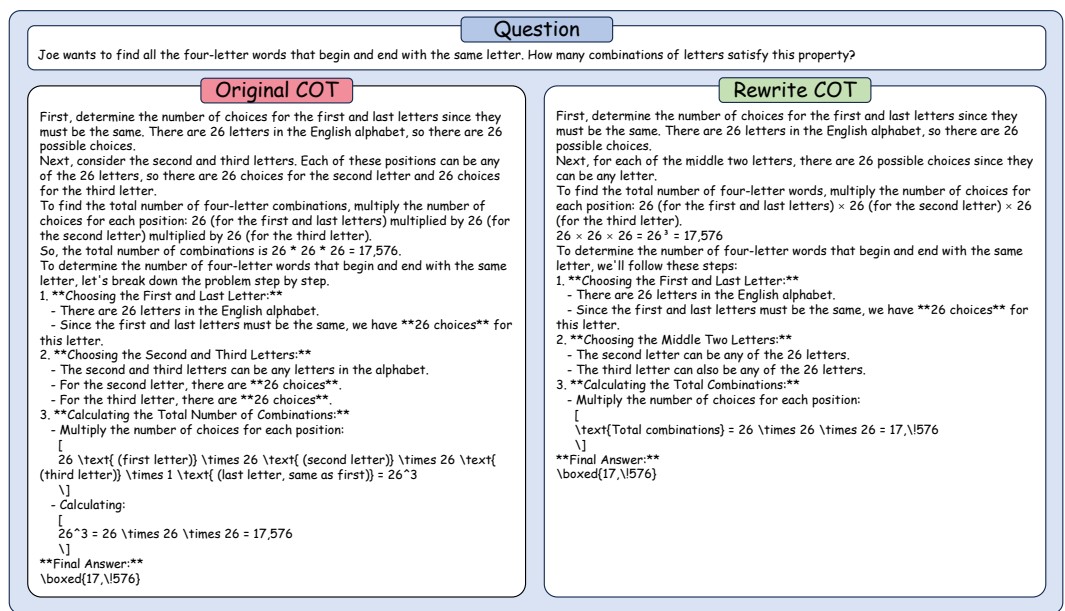

Figure 6: Reasoning trace comparison for a constrained letter-counting problem before and after refinement.

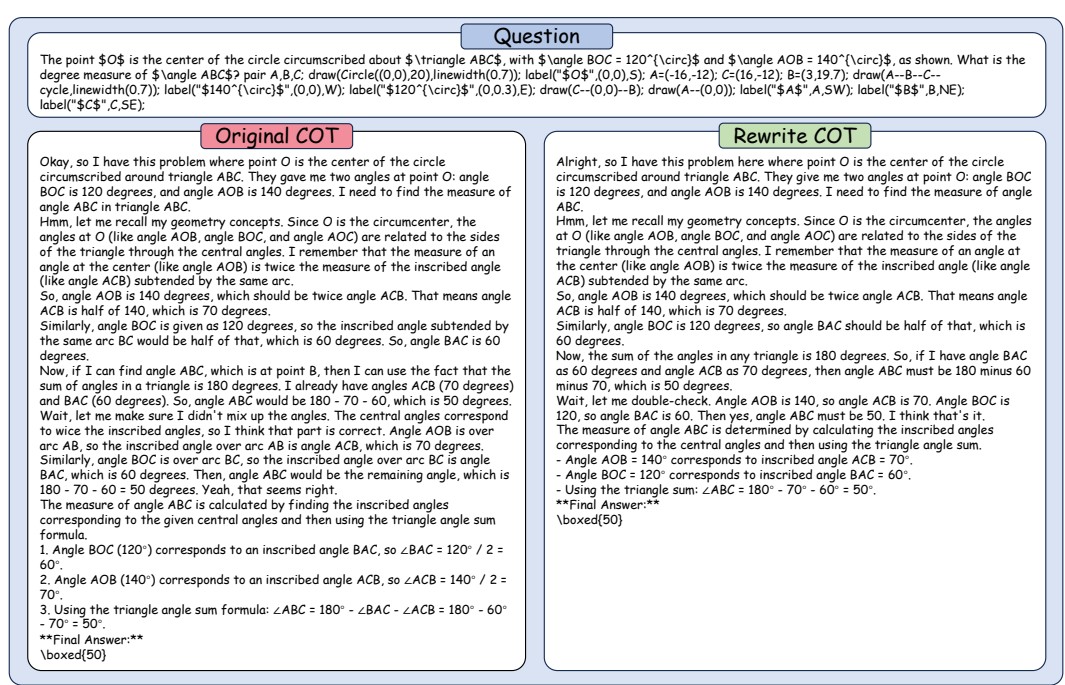

Figure 7: Reasoning trace comparison for a circle geometry problem before and after refinement.

