# OpenReview forum: "TokenSqueeze: Performance-Preserving Compression for Reasoning LLMs"
_NeurIPS.cc/2025/Conference — NeurIPS 2025 poster_

### Official Review · Reviewer_mo3k · 2025-06-23

**Clarity:** 3
**Significance:** 3
**Originality:** 3
**Rating:** 4
**Confidence:** 4

**Summary:**

The paper introduces TokenSqueeze, a novel method for compressing reasoning traces in large language models (LLMs) to improve reasoning efficiency without sacrificing accuracy. TokenSqueeze addresses the issue of long chain-of-thought (CoT) traces in reasoning LLMs, which lead to increased inference latency and memory consumption. The method includes three main components: adaptive reasoning depth selection to maintain appropriate reasoning depth based on problem complexity, intra-step linguistic refinement to enhance the conciseness of reasoning steps while preserving their meaning, and a composite optimization objective that balances correctness and brevity. Through comprehensive experiments on mathematical reasoning and programming tasks, TokenSqueeze demonstrates significant reductions in token usage while maintaining or improving accuracy, achieving higher efficiency and better performance under token budget constraints compared to existing methods.

**Questions:**

See weakness.

**Ethical Concerns:**

["NO or VERY MINOR ethics concerns only"]

**Final Justification:**

Thanks for the rebuttal. All of my concerns have been addressed. I have updated my score to Borderline accept.

**Limitations:**

No.

**Paper Formatting Concerns:**

No.

**Quality:**

3

**Strengths And Weaknesses:**

Strengths：
1. The method for constructing contrastive preference data from self-generated reasoning traces is original and effective.
2. The core idea of optimizing reasoning efficiency via length-aware preference modeling is reasonable and addresses a practical need.
3. The paper is well-structured and easy to follow.

Weaknesses：
1. While the proposed DPO-L formulation integrates response length into the preference loss, the core idea of using length-based regularization in preference optimization has been explored in prior works.[1,2]
2. The formulation in Equation (2) lacks clarity. Specifically, it is unclear whether the \( j \)-th token refers to the position following the \( i \)-th reasoning step, as implied in the main text. A more precise definition would improve interpretability.
3. The paper would benefit from including experiments with alternative backbone models to verify the generalizability of the proposed method across different architectures.



[1] O1-Pruner: Length-Harmonizing Fine-Tuning for O1-Like Reasoning Pruning 2025.
[2] Kimi k1.5: Scaling Reinforcement Learning with LLMs 2025.

---

> ### Author Rebuttal · Authors · 2025-07-31
>
> We sincerely thank you for your careful review and positive feedback. We appreciate your recognition of the originality of our approach, the practicality of our objective, and the overall clarity of the paper. Your encouraging comments are very helpful to us, and we address the remaining points in detail below.
>
> **W1:** While the proposed DPO-L formulation integrates response length into the preference loss, the core idea of using length-based regularization in preference optimization has been explored in prior works.[1,2]
>
> **Response:** In contrast to approaches like *O1-Pruner* and *Kimi-k1.5*, which primarily rely on adding a length regularization term within a policy gradient framework, our core idea is not simply to impose a regularizer. Instead, we propose a systematic approach to preference optimization that improves inference efficiency **without sacrificing correctness**. Our method enables the model to implicitly prefer responses that are both accurate and concise, offering a fundamentally different paradigm from methods that focus solely on length penalties.
>
> Moreover, the way training data is constructed also differs fundamentally. We automatically derive $(\mathrm{chosen\ sample}, \mathrm{rejected\ sample})$ pairs from model outputs: for each prompt, the “chosen” response is correct and shorter, while longer, incorrect ones are labeled as “rejected.” In contrast, *O1-Pruner* and *Kimi-k1.5* construct their training data as $(\mathrm{sample}, \mathrm{reward})$ pairs, where each response is assigned a scalar reward.
>
> **W2:** The formulation in Equation (2) lacks clarity. Specifically, it is unclear whether the ( j )-th token refers to the position following the ( i )-th reasoning step, as implied in the main text. A more precise definition would improve interpretability.
>
> **Response:** We appreciate your comment and will clarify this in the camera-ready version if the paper is accepted. Specifically, in Equation $2$:
> $$
> D_{\mathrm{KL}}^{\text{full}}
> \approx
> \sum_{j=1}^{\min(T,L)}
> D_{\mathrm{KL}}\left(
> Q_{\theta}(\cdot \mid p, s_{\leq i}, t_{1:j-1}) \,\|\, Q_{\theta}(\cdot \mid p, s_{<i}, s_i', t_{1:j-1})
> \right),
> $$
>
> We clarify that $t_j$ refers to the $j$-th token in the consecutive reasoning steps following the $i$-th step $s_i$. Specifically, $t_1$ corresponds to the first token of $s_{i+1}$. If the tokens in $s_{i+1}$ are insufficient to fill the window, we continue accumulating tokens from subsequent steps until the window is filled or the response ends.
>
> To provide more insight into Equation 2, we provide the derivation from Equation 1 to Equation 2 below.
>
> **Proof (from Equation (1) to Equation (2))**
>
> ---
>
> **Notation and Setup**
>
> We begin by stating the two key KL divergence expressions from the paper:
>
> - Equation (1):
>   $$
>   D_{\mathrm{KL}}\left( P_{\theta}(\cdot \mid p, s_{\leq i}) \| P_{\theta}(\cdot \mid p, s_{<i}, s_i') \right)
>   $$
>
> - Equation (2):
>   $$
>   D_{\mathrm{KL}}^{\text{full}}
>   \approx
>   \sum_{j=1}^{\min(T,L)}
>   D_{\mathrm{KL}}\left(
>   Q_{\theta}(\cdot \mid p, s_{\leq i}, t_{1:j-1}) \| Q_{\theta}(\cdot \mid p, s_{<i}, s_i', t_{1:j-1})
>   \right)
>   $$
>
> These expressions measure the change in predicted continuation distributions before and after rewriting step $s_i$.
>
> We now define the key variables and distributions involved in the KL computation:
>
> - $p$: the input prompt to the model.
> - $s_i$: the $i$‑th reasoning step generated by the model.
> - $s_{\le i}$: the reasoning trace up to and including step $i$ (i.e., $s_1, s_2, \dots, s_i$).
> - $s'_i$: a rewritten version of step $s_i$ used in intra-step refinement.
> - $t_j$: the $j$‑th token **after** $s_i$ (or $s'_i$), starting the continuation sequence.
> - $t_{1:T}$: a full continuation sequence of $T$ tokens (including EOS).
>
> We define the following probability distributions:
>
> - $Q_\theta(\cdot \mid \cdot)$: the model’s **next-token distribution** under the given context (token-level), parameterized by $\theta$.
> - $P_\theta(\cdot \mid \cdot)$: the model’s **sequence-level distribution** over continuations $t_{1:T}$, given the input context.
>
> To simplify notation, we define:
>
> - $\mathsf{A}(t_{1:T}) := P_\theta(t_{1:T} \mid p, s_{\le i})$
> - $\mathsf{B}(t_{1:T}) := P_\theta(t_{1:T} \mid p, s_{< i}, s'_i)$
>
> These represent the two sequence distributions compared in the KL divergence.
>
> ---
>
> **Step 1: Sequence‑Level KL Definition**
>
> The KL term in Equation (1) is:
> $$
> D\_{\mathrm{KL}}\big( \mathsf{A} \| \mathsf{B} \big)=\mathbb{E}\_{t\_{1:T} \sim \mathsf{A}}
> \left[
> \log \frac{ \mathsf{A}(t\_{1:T}) }{ \mathsf{B}(t\_{1:T}) }
> \right].
> $$
>
> Expanding both $\mathsf{A}$ and $\mathsf{B}$ autoregressively using $Q_\theta$:
> $$
> \begin{aligned}
> \mathsf{A}(t\_{1:T}) = \prod\_{j=1}^{T} Q\_\theta(t\_j \mid p, s\_{\le i}, t\_{1:j-1}), \\
> \mathsf{B}(t\_{1:T}) = \prod\_{j=1}^{T} Q\_\theta(t\_j \mid p, s\_{< i}, s'\_i, t\_{1:j-1}).
> \end{aligned}
> $$
>
> Substituting into the KL definition:
> $$
> \begin{aligned}
> D\_{\mathrm{KL}}(\mathsf{A} \| \mathsf{B})
> &= \mathbb{E}\_{t\_{1:T} \sim \mathsf{A}}
> \left[
> \sum\_{j=1}^{T}
> \log
> \frac{
> Q\_\theta(t\_j \mid p, s\_{\le i}, t\_{1:j-1})
> }{
> Q\_\theta(t\_j \mid p, s\_{< i}, s'\_i, t\_{1:j-1})
> }
> \right] \\
> = \sum\_{j=1}^{T}
> \mathbb{E}\_{t\_{1:j-1} \sim \mathsf{A}}
> \left[
> D\_{\mathrm{KL}}\left(
> Q\_\theta(\cdot \mid p, s\_{\le i}, t\_{1:j-1})
> \Big\|
> Q\_\theta(\cdot \mid p, s\_{< i}, s'\_i, t\_{1:j-1})
> \right)
> \right].
> \end{aligned}
> $$
>
> This shows that the **sequence-level KL** equals the expected sum of per-token conditional KL divergences.
>
> ---
>
> **Step 2: Monte Carlo Approximation**
>
> Computing the full expectation over all possible token histories $t_{1:j-1}$ is intractable in practice. To address this, we use a Monte Carlo approximation: instead of averaging over all continuations, we compute the KL terms along an existing sampled trajectory $t_{1:T}$ drawn from $\mathsf{A}$:
> $$
> D\_{\mathrm{KL}}^{\text{full}}
> \approx
> \sum\_{j=1}^{T}
> D\_{\mathrm{KL}}\left(
> Q\_\theta(\cdot \mid p, s\_{\le i}, t\_{1:j-1})
> \Big\|
> Q\_\theta(\cdot \mid p, s\_{< i}, s'\_i, t\_{1:j-1})
> \right).
> $$
>
> ---
>
> **Step 3: Token-Window Truncation**
>
> To reduce cost further, we truncate the sum at a fixed window size $L$ (e.g., 512 tokens):
> $$
> D\_{\mathrm{KL}}^{\text{full}}
> \approx
> \sum\_{j=1}^{\min(T, L)}
> D\_{\mathrm{KL}}\left(
> Q\_\theta(\cdot \mid p, s\_{\le i}, t\_{1:j-1})
> \Big\|
> Q\_\theta(\cdot \mid p, s\_{< i}, s'\_i, t\_{1:j-1})
> \right).
> $$
>
> This is exactly the formulation given in **Equation (2)** of the main paper.
>
> The approximation introduces a residual term:
> $$
> R\_L = \sum\_{j > L}
> \mathbb{E}\_{t\_{1:j-1} \sim \mathsf{A}} \left[
> D\_{\mathrm{KL}}(\cdot \| \cdot)
> \right] \ge 0,
> $$
> which is small when the effect of rewriting $s_i$ fades over time.
>
>
> **W3:** The paper would benefit from including experiments with alternative backbone models to verify the generalizability of the proposed method across different architectures.
>
> **Response:** To further demonstrate the effectiveness of our model across different backbones and LLM scales, we conducted experiments using Phi4-Reasoning-14B. The results are presented below:
>
> | Model               | **Dataset** | **Accuracy (%)** | **AUC**  | **Avg. Length** |
> | ------------------- | ----------- | ---------------- | -------- | --------------- |
> | Phi4-Reasoning-14B  | AIME24      | 72.1             | 55.1     | 7768            |
> | TokenSqueeze (Phi4) | AIME24      | **73.8**         | **59.6** | **6326**        |
> | Phi4-Reasoning-14B  | MATH500     | 93.8             | 72.0     | 7644            |
> | TokenSqueeze (Phi4) | MATH500     | **96.0**         | **87.5** | **2921**        |
>
> It is important to note that, due to time constraints, we reducing K in intra-step linguistic refinement from 64 to 16. We expect that further hyperparameter tuning would yield even better performance.

---

### Official Review · Reviewer_3A86 · 2025-06-30

**Clarity:** 3
**Significance:** 3
**Originality:** 3
**Rating:** 4
**Confidence:** 3

**Summary:**

TokenSqueeze proposes a training-time method to compress reasoning traces in LLMs via three innovations: (1) adaptive reasoning depth selection to retain essential steps based on problem difficulty, (2) KL-constrained intra-step linguistic refinement to boost conciseness without altering semantics, and (3) a length-aware preference objective (DPO-L) that jointly optimizes correctness and brevity. Evaluated on math (MATH500, AIME) and coding (LiveCodeBench) benchmarks, TokenSqueeze reduces token usage by 20–51% while preserving accuracy, outperforming methods like DAST and Kimi-k1.5.

**Questions:**

See the Weakness.

**Ethical Concerns:**

["NO or VERY MINOR ethics concerns only"]

**Final Justification:**

The detailed response solves my concern.

**Limitations:**

yes

**Paper Formatting Concerns:**

N/A.

**Quality:**

3

**Strengths And Weaknesses:**

Strength

- Intra-step refinement under KL constraints ensures semantic fidelity while shortening steps, a significant improvement over rule-based truncation.

- 50% average token reduction on MATH500 with negligible accuracy drop (92.8% → 92.4%) for DeepSeek-R1-7B.

- Validated across model sizes (1.5B/7B) and tasks (math/code), demonstrating broad applicability.

Weakness

- Intra-step refinement requires resampling K=64 candidates per step and KL calculations over token windows (Eq. 2), adding significant preprocessing costs. Training efficiency claims lack quantification (e.g., GPU hours vs. baseline).

- Performance hinges on α (depth selection) and ε (KL threshold). No analysis explores how ε affects the brevity-fidelity trade-off.

- The KL approximation (Eq. 2) lacks formal justification for preserving global reasoning integrity.

---

> ### Author Rebuttal · Authors · 2025-07-30
>
> We sincerely thank you for your careful review and positive assessment of our work. Your feedback is greatly valued, and we respond to your points in detail below.
>
> **Response to W1:** We conducted an experiment using 1,000 randomly sampled initial question prompts on a single node with 8×A100 GPUs. The runtime (in hours) for each major preprocessing stage is shown below:
>
> | Stage                         | Time (hours) |
> | - | - |
> | Data Sampling                 | 1.73         |
> | Intra-step Candidate Sampling | 3.91         |
> | KL Divergence Computation     | 3.23         |
>
> This runtime is comparable to standard preprocessing pipelines in industrial practice, such as reward-based rejection sampling, and remains within acceptable bounds for large-scale training. Moreover, since our method constructs the dataset offline, it incurs this cost only once and can be reused across multiple training runs, unlike online reinforcement learning methods that require repeated sampling during training.
>
> To further reduce computational cost, we have implemented several optimization strategies:
>
> - **KV-cache reuse:**
>   When processing step $s_{i+1}$, we reuse the key-value cache computed during the sampling of step $s_i$ in both resampling and KL divergence calculation, significantly reducing the cost of autoregressive decoding.
>
> - **Selective rewriting:**
>   We first filter generated samples to identify preference-worthy candidates (e.g., concise and correct answers) before rewriting. This ensures rewriting is applied only to a targeted subset.
>
> - **KL window approximation:**
>   The KL divergence is approximated within a fixed-length window of 512 tokens:
>
>   $$
>   D_{\mathrm{KL}}^{\text{full}}
>   \approx
>   \sum_{j=1}^{\min(T, L)} D_{\mathrm{KL}}\left( Q_{\theta}(\cdot \mid p, s_{\leq i}, t_{1:j-1}) \,\|\, Q_{\theta}(\cdot \mid p, s_{<i}, s_i', t_{1:j-1}) \right),
>   $$
>
>   This avoids full-sequence computation and offers significant resource savings.
>
> - **Choice of $K$:**
>   We would like to clarify that the resampling parameter $K = 64$ was chosen as a strong setting to push performance to its limit for benchmark comparison. However, our empirical observations indicate that smaller values such as $K = 32$ still achieve highly competitive performance while significantly reducing computational cost. This suggests that such a large value is not strictly necessary in practice. We believe that a more comprehensive investigation of the impact of different $K$ values is an important direction for future work, and we plan to explore this systematically in subsequent versions of the paper or follow-up research. As shown in our response to Reviewer 1 (W1), the experiments on Phi-4-Reasoning-14B with $K = 16$ still demonstrate a significant improvement in the model’s efficiency.
>
>
>
> **Response to W2:** As shown in Figure 4 of the paper, we analyzed the performance impact of different $\alpha$ values and found that extreme values (too large or too small) negatively affect both accuracy and token efficiency. We selected $\alpha = 0.2$ as it achieves the best balance between brevity and correctness.
>
> For the KL threshold $\varepsilon$, we selected its value based on empirical observations. To further investigate the effect of $\varepsilon$, we additionally conducted a quantitative experiment with $\varepsilon = 0.02$. The results are shown below:
>
> |                     | AIME24 Acc | AIME24 AUC | AIME24 Len | MATH500 Acc | MATH500 AUC | MATH500 Len |
> | -- | ---------- | ------ | ------ | ----------- | ----------- | ----------- |
> | Baseline            | 55.5       | 41.6       | 7542       | 92.8        | 83.6        | 3638        |
> | $\varepsilon=0.02$  | 46.9       | 39.7       | 5019       | 88.0        | 83.5        | 1695        |
> | $\varepsilon=0.005$ | 57.5       | 48.5       | 5157       | 92.4        | 87.5        | 1773        |
>
> These results confirm that excessively high values of $\varepsilon$ can negatively affect accuracy. We plan to explore the effect of varying $\varepsilon$ more systematically in future work.
>
> In addition to the quantitative results, we also provide a qualitative example that illustrates how different values of $\varepsilon$ influence the form and length of the generated rewrites:
>
> |                  | Original Text     | $\varepsilon=0.005$    | $\varepsilon=0.01$    | $\varepsilon=0.02$     |
> | ----- | ---------- | ----- | ---------- | ------ |
> | **Text**         | Now, since each test score is an integer between 0 and 100, inclusive, both $x$ and $y$ must be integers in that range. Also, since $y$ is higher than $x$, $x$ must be at least 0 and $y$ could be up to 100. | Now, since each test score is an integer between 0 and 100, inclusive, both $x$ and $y$ have to be integers in that range. Also, $y$ must be greater than $x$. | Also, since each test score is an integer between 0 and 100, inclusive, both $x$ and $y$ must be integers within that range. | Also, both $x$ and $y$ are integers between 0 and 100, inclusive. |
> | **Token Length** | 75   | 52    | 37     | 25    |
>
> **Response to W3:** Thank you for your question. Due to space limitations, we could not include the full derivation in the main paper. If the paper is accepted, we will provide the complete proof in the appendix of the camera-ready version.
>
> Since both data generation and step rewriting are performed on the same model, we define the original step $s_i$ and its refined version $s_i'$ as semantically equivalent if they satisfy the following condition:
> $$
> D_{\mathrm{KL}}\left( P_{\theta}(\cdot \mid p, s_{\leq i}) \,\|\, P_{\theta}(\cdot \mid p, s_{<i}, s_i') \right) < \varepsilon,
> $$
> as formulated in Equation (1). This constraint ensures that rewriting does not compromise the semantic integrity of the reasoning trace.
>
> Below we provide the derivation from Equation (1) to Equation (2):
>
> **Proof (from Equation (1) to Equation (2))**
>
> ---
>
> **Notation and Setup**
>
> We begin by stating the two key KL divergence expressions from the paper:
>
> - Equation (1):
>   $$
>   D_{\mathrm{KL}}\left( P_{\theta}(\cdot \mid p, s_{\leq i}) \| P_{\theta}(\cdot \mid p, s_{<i}, s_i') \right)
>   $$
>
> - Equation (2):
>   $$
>   D_{\mathrm{KL}}^{\text{full}}
>   \approx
>   \sum_{j=1}^{\min(T,L)}
>   D_{\mathrm{KL}}\left(
>   Q_{\theta}(\cdot \mid p, s_{\leq i}, t_{1:j-1}) \| Q_{\theta}(\cdot \mid p, s_{<i}, s_i', t_{1:j-1})
>   \right)
>   $$
>
> These expressions measure the change in predicted continuation distributions before and after rewriting step $s_i$.
>
> We now define the key variables and distributions involved in the KL computation:
>
> - $p$: the input prompt to the model.
> - $s_i$: the $i$‑th reasoning step generated by the model.
> - $s_{\le i}$: the reasoning trace up to and including step $i$ (i.e., $s_1, s_2, \dots, s_i$).
> - $s'_i$: a rewritten version of step $s_i$ used in intra-step refinement.
> - $t_j$: the $j$‑th token **after** $s_i$ (or $s'_i$), starting the continuation sequence.
> - $t_{1:T}$: a full continuation sequence of $T$ tokens (including EOS).
>
> We define the following probability distributions:
>
> - $Q_\theta(\cdot \mid \cdot)$: the model’s **next-token distribution** under the given context (token-level), parameterized by $\theta$.
> - $P_\theta(\cdot \mid \cdot)$: the model’s **sequence-level distribution** over continuations $t_{1:T}$, given the input context.
>
> To simplify notation, we define:
>
> - $\mathsf{A}(t_{1:T}) := P_\theta(t_{1:T} \mid p, s_{\le i})$
> - $\mathsf{B}(t_{1:T}) := P_\theta(t_{1:T} \mid p, s_{< i}, s'_i)$
>
> These represent the two sequence distributions compared in the KL divergence.
>
> ---
>
> **Step 1: Sequence‑Level KL Definition**
>
> The KL term in Equation (1) is:
> $$
> D\_{\mathrm{KL}}\big( \mathsf{A} \| \mathsf{B} \big)=\mathbb{E}\_{t\_{1:T} \sim \mathsf{A}}
> \left[
> \log \frac{ \mathsf{A}(t\_{1:T}) }{ \mathsf{B}(t\_{1:T}) }
> \right].
> $$
>
> Expanding both $\mathsf{A}$ and $\mathsf{B}$ autoregressively using $Q_\theta$:
> $$
> \begin{aligned}
> \mathsf{A}(t\_{1:T}) = \prod\_{j=1}^{T} Q\_\theta(t\_j \mid p, s\_{\le i}, t\_{1:j-1}), \\
> \mathsf{B}(t\_{1:T}) = \prod\_{j=1}^{T} Q\_\theta(t\_j \mid p, s\_{< i}, s'\_i, t\_{1:j-1}).
> \end{aligned}
> $$
>
> Substituting into the KL definition:
> $$
> \begin{aligned}
> D\_{\mathrm{KL}}(\mathsf{A} \| \mathsf{B})
> &= \mathbb{E}\_{t\_{1:T} \sim \mathsf{A}}
> \left[
> \sum\_{j=1}^{T}
> \log
> \frac{
> Q\_\theta(t\_j \mid p, s\_{\le i}, t\_{1:j-1})
> }{
> Q\_\theta(t\_j \mid p, s\_{< i}, s'\_i, t\_{1:j-1})
> }
> \right] \\
> = \sum\_{j=1}^{T}
> \mathbb{E}\_{t\_{1:j-1} \sim \mathsf{A}}
> \left[
> D\_{\mathrm{KL}}\left(
> Q\_\theta(\cdot \mid p, s\_{\le i}, t\_{1:j-1})
> \Big\|
> Q\_\theta(\cdot \mid p, s\_{< i}, s'\_i, t\_{1:j-1})
> \right)
> \right].
> \end{aligned}
> $$
>
> This shows that the **sequence-level KL** equals the expected sum of per-token conditional KL divergences.
>
> ---
>
> **Step 2: Monte Carlo Approximation**
>
> Computing the full expectation over all possible token histories $t_{1:j-1}$ is intractable in practice. To address this, we use a Monte Carlo approximation: instead of averaging over all continuations, we compute the KL terms along an existing sampled trajectory $t_{1:T}$ drawn from $\mathsf{A}$:
> $$
> D\_{\mathrm{KL}}^{\text{full}}
> \approx
> \sum\_{j=1}^{T}
> D\_{\mathrm{KL}}\left(
> Q\_\theta(\cdot \mid p, s\_{\le i}, t\_{1:j-1})
> \Big\|
> Q\_\theta(\cdot \mid p, s\_{< i}, s'\_i, t\_{1:j-1})
> \right).
> $$
>
> ---
>
> **Step 3: Token-Window Truncation**
>
> To reduce cost further, we truncate the sum at a fixed window size $L$ (e.g., 512 tokens):
> $$
> D\_{\mathrm{KL}}^{\text{full}}
> \approx
> \sum\_{j=1}^{\min(T, L)}
> D\_{\mathrm{KL}}\left(
> Q\_\theta(\cdot \mid p, s\_{\le i}, t\_{1:j-1})
> \Big\|
> Q\_\theta(\cdot \mid p, s\_{< i}, s'\_i, t\_{1:j-1})
> \right).
> $$
>
> This is exactly the formulation given in **Equation (2)** of the main paper.
>
> The approximation introduces a residual term:
> $$
> R\_L = \sum\_{j > L}
> \mathbb{E}\_{t\_{1:j-1} \sim \mathsf{A}} \left[
> D\_{\mathrm{KL}}(\cdot \| \cdot)
> \right] \ge 0,
> $$
> which is small when the effect of rewriting $s_i$ fades over time.

---

> > ### Comment · Reviewer_3A86 · 2025-08-07
> >
> > Thank the authors for the detailed response which solves my concern. I keep my positive score.

---

### Official Review · Reviewer_Cefq · 2025-06-30

**Clarity:** 3
**Significance:** 3
**Originality:** 3
**Rating:** 4
**Confidence:** 4

**Summary:**

Reasoning LLMs such as OpenAI-o1 and DeepSeek-R1 attain strong performance by producing CoT reasoning traces, but these verbose outputs incur high latency and memory costs. TokenSqueeze addresses this efficiency–accuracy trade-off via a self-supervised, three-stage Long2Short approach: (1) adaptive depth selection dynamically filters reasoning paths based on problem difficulty; (2) intra-step linguistic refinement rewrites each step under a KL-divergence constraint to maximize information density; and (3) a length-aware preference objective fine-tunes the model to balance correctness with shortness.

**Questions:**

As I mentioned in weakness:

1. The generated reasoning lengths appear unstable. Were multiple random seeds used in the experiments to evaluate robustness?
2. It would strengthen the paper to compare results with state-of-the-art methods such as O1-Pruner, L1, and others.

**Ethical Concerns:**

["NO or VERY MINOR ethics concerns only"]

**Final Justification:**

The authors addressed my concerns, and I would like to keep the positive rating.

**Limitations:**

yes

**Quality:**

3

**Strengths And Weaknesses:**

**Strengths:**

1. The paper is well-written and generally easy to follow.
2. The second idea—**intra-step refinement**—is particularly interesting.
3. The experimental results demonstrate strong performance, successfully reducing token length while maintaining accuracy.

**Weaknesses:**

1. In line 13, the authors mention "we propose to selectively identify self-generated samples." While the general concept becomes clearer in subsequent sections, this point is initially vague and could benefit from clarification upfront.
2. In line 48, the paper states that when the reasoning length exceeds a certain token threshold, the correlation between token count and model performance significantly weakens. This observation would be more appropriately framed as part of the motivation in the Methods section rather than being introduced in the Experiments section.
3. The paper lacks an ablation study on the choice of \( M \), as referenced in line 139. This analysis is important for understanding the impact of the selection strategy.
4. The notation \( s \) is used inconsistently: in line 135 it refers to reasoning traces, while in line 151 it denotes a reasoning step. This dual usage creates confusion and should be clarified.
5. The symbol \( \alpha \) appears twice with different meanings (e.g., line 132 and Equation 4). These should be clearly distinguished or redefined to avoid ambiguity.
6. In line 137, it is unclear how the "longer incorrect response" is obtained. More detail on this process would be helpful.
7. In line 157, the paper states that the level of information preservation is "tunable." Is this tunability controlled manually, or is it learned during training? This should be specified.
8. In Section 3.2, the method for accurately segmenting a reasoning step is unclear, especially since reasoning steps may contain many tokens and ambiguous boundaries. How is the stopping criterion determined?
9. The generated reasoning lengths appear unstable. Were multiple random seeds used in the experiments to evaluate robustness?
10. It would strengthen the paper to compare results with state-of-the-art methods such as O1-Pruner, L1, and others.

[1] O1-pruner: Length-harmonizing fine-tuning for o1-like reasoning pruning.

[2] L1: Controlling how long a reasoning model thinks with reinforcement learning.

---

> ### Author Rebuttal · Authors · 2025-07-31
>
> We sincerely thank you for your careful reading and constructive feedback. We appreciate your recognition of the overall clarity of the paper, the novelty of the intra-step refinement component, and the effectiveness of our method in reducing token usage while preserving accuracy. Your comments have helped us identify several areas for clarification and improvement, and we address each of your points in detail below.
>
> **W1 W2 W3 W4:**
>
> - In line 13, the authors mention "we propose to selectively identify self-generated samples." While the general concept becomes clearer in subsequent sections, this point is initially vague and could benefit from clarification upfront.
>
> - In line 48, the paper states that when the reasoning length exceeds a certain token threshold, the correlation between token count and model performance significantly weakens. This observation would be more appropriately framed as part of the motivation in the Methods section rather than being introduced in the Experiments section.
>
> - The notation $s$ is used inconsistently: in line 135 it refers to reasoning traces, while in line 151 it denotes a reasoning step. This dual usage creates confusion and should be clarified.
>
> - The symbol $\alpha$appears twice with different meanings (e.g., line 132 and Equation 4). These should be clearly distinguished or redefined to avoid ambiguity.
>
> **Response:** Thank you for pointing out these important issues. If the paper is accepted, we will make the following revisions in the camera-ready version:
>
> 1. In the abstract, we will revise the sentence
>    *“we propose to selectively identify self-generated samples that preserve the model’s competency while minimizing token usage”*
>    to
>    *“we propose to select self-generated samples whose reasoning depth is adaptively matched to the complexity of the problem”*
>    to improve clarity.
>
> 2. In lines 132 and 135, we will revise the overloaded symbols used for quantile computation and reasoning trace collections to eliminate ambiguity and improve readability.
>
> 3. Regarding the placement of the token-threshold observation (line 48), we acknowledge your suggestion and will revise it accordingly in the camera-ready version if the paper is accepted. Due to space limitations, we omit the details here.
>
> **W3:** The paper lacks an ablation study on the choice of $M$, as referenced in line 139. This analysis is important for understanding the impact of the selection strategy.
>
> **Response:** To further investigate the impact of the number of samples per instance $M$ during the data collection phase, we additionally trained DeepSeek-R1-Distill-Qwen-7B with $M = 32$. The results are presented below, where *Baseline* refers to the original unmodified DeepSeek-R1-Distill-Qwen-7B model:
>
> |          | AIME24 Acc | AIME24 AUC | AIME24 Len | MATH500 Acc | MATH500 AUC | MATH500 Len |
> | -------- | ---------- | ---------- | ---------- | ----------- | ----------- | ----------- |
> | Baseline | 55.5       | 41.6       | 7542       | 92.8        | 83.6        | 3638        |
> | $M = 32$ | 56.5       | 46.8       | 5596       | 92.4        | 87.1        | 1875        |
> | $M = 64$ | 57.5       | 48.5       | 5157       | 92.4        | 87.5        | 1773        |
>
> These additional experiments allow us to assess how varying the sampling number $M$ influences the trade-off between reasoning performance and inference efficiency.
>
> **W6:** In line 137, it is unclear how the "longer incorrect response" is obtained. More detail on this process would be helpful.
>
> **Response:** The “long and incorrect” samples are defined relative to the chosen ones. Specifically, during data construction, for each chosen sample, we select rejected samples that satisfy both of the following conditions:
>
> 1. The response is longer than the chosen sample in token length.
> 2. The response is classified as incorrect based on final answer comparison.
>
> This design ensures that the preference objective simultaneously penalizes both verbosity and incorrectness, aligning the optimization with our intended trade-off between accuracy and efficiency.
>
> **W7:** In line 157, the paper states that the level of information preservation is "tunable." Is this tunability controlled manually, or is it learned during training? This should be specified.
>
> **Response:** The tunability of information preservation is implemented at **training time during data preprocessing**.
>
> Our method allows adjusting the strictness of information preservation in the rewriting phase via a KL divergence threshold $\varepsilon$. This hyperparameter governs how much semantic deviation is tolerated when rewriting each reasoning step. Specifically, given a set of candidate rewrites $\{s_i^{(k)}\}$, we select the shortest one that satisfies the following constraint:
> $$
> \min_{s_i' \in \{s_i^{(k)}\}} \ell(s_i') \quad \text{subject to} \quad D_{\mathrm{KL}}\left( P_{\theta}(\cdot \mid p, s_{\leq i}) \,\|\, P_{\theta}(\cdot \mid p, s_{<i}, s_i') \right) < \varepsilon
> $$
> Here, $\ell(s_i')$ denotes the token length of the rewritten step, and $P_\theta$ is the model's sequence-level distribution. This KL-based constraint ensures that the semantic integrity of the original reasoning trace is preserved during the offline data refinement stage.
>
> Below, we provide a concrete example illustrating how different threshold values affect the resulting rewrites and their lengths:
>
> |                  | Original Text                                                | $\varepsilon=0.005$                                          | $\varepsilon=0.01$                                           | $\varepsilon=0.02$                                           |
> | :--------------: | :----------------------------------------------------------- | :----------------------------------------------------------- | :----------------------------------------------------------- | :----------------------------------------------------------- |
> |     **Text**     | Now, since each test score is an integer between 0 and 100, inclusive, both $x$ and $y$ must be integers in that range. Also, since $y$ is higher than $x$, $x$ must be at least 0 and $y$ could be up to 100. | Now, since each test score is an integer between 0 and 100, inclusive, both $x$ and $y$ have to be integers in that range. Also, $y$ must be greater than $x$. | Also, since each test score is an integer between 0 and 100, inclusive, both $x$ and $y$ must be integers within that range. | Also, both $x$ and $y$ are integers between 0 and 100, inclusive. |
> | **Token Length** | 75                                                           | 52                                                           | 37                                                           | 25                                                           |
>
> **W8:** In Section 3.2, the method for accurately segmenting a reasoning step is unclear, especially since reasoning steps may contain many tokens and ambiguous boundaries. How is the stopping criterion determined?
>
> **Response:** We segment reasoning steps based on **double newline characters `\n\n`**, which is a common convention adopted by most mainstream open-source LLMs. This formatting is naturally used by model family such as DeepSeeks and Qwens when generating step-by-step reasoning outputs. As a result, our segmentation aligns well with model behavior and does not require additional annotation or post-processing.
>
> **W9:** The generated reasoning lengths appear unstable. Were multiple random seeds used in the experiments to evaluate robustness?
>
> **Response:** Thank you for raising this point. For smaller datasets such as AIME24 and AIME25, we adopted an evaluation strategy designed to ensure robustness against potential variability in response lengths. Specifically, we averaged the results over 16 independent inference runs to obtain stable performance estimates. All generations were conducted using the default random seed provided by the vLLM framework.
>
> **W10:** It would strengthen the paper to compare results with state-of-the-art methods such as O1-Pruner, L1, and others.
>
> **Response:** Since O1-Pruner and L1 are built upon different base models than ours, their results are not directly comparable to our setting. Therefore, in the paper, we compared our method with DAST and TrainEffi, which utilize the same base model.
>
> Considering that L1 focuses more on *controllable reasoning length*, while O1-Pruner emphasizes *reasoning efficiency*, we additionally reproduced O1-Pruner using its open-sourced codebase. Following the official configuration of O1-Pruner, we reproduced their method on a single node with 8 × A100 80GB GPUs, using DeepSeek-R1-Distill-Qwen-7B as the base model. We strictly followed the hyperparameter settings described in the original paper, without any modifications.
>
> The reproduction results are summarized below:
>
> |           | AIME24 Acc | AIME24 AUC | AIME24 Len | MATH500 Acc | MATH500 AUC | MATH500 Len |
> | --------- | ---------- | ---------- | ---------- | ----------- | ----------- | ----------- |
> | Baseline  | 55.5       | 41.6       | 7542       | 92.8        | 83.6        | 3638        |
> | O1-Pruner | 51.0       | 41.3       | 6281       | **93.0**    | 85.2        | 2764        |
> | Ours      | **57.5**   | **48.5**   | **5157**   | 92.4        | **87.5**    | **1773**    |
>
> As shown in the results, our method achieves significantly higher accuracy on challenging tasks like AIME24, while also delivering superior AUC and substantially reduced reasoning lengths compared to O1-Pruner.

---

> ### Comment · Reviewer_Cefq · 2025-08-05
>
> The authors addressed my concerns, and I would like to keep the positive rating.

---

### Official Review · Reviewer_JKMo · 2025-07-07

**Clarity:** 4
**Significance:** 3
**Originality:** 3
**Rating:** 5
**Confidence:** 4

**Summary:**

This paper proposes *TokenSqueeze*, a method to compress the reasoning traces by choosing rewrites that would minimize the length of while subject to a KL-divergence constraint of the token distributions of the next reasoning steps. It then uses the chosen shorter reasoning trace to train the model using a modified version of the DPO algorithm that also incorporates the length preference. Experiments are conducted on AIME, MATH, LiveCodeBench datasets and the proposed methods are compared with very recent baselines such as DAST and TrainEffi. Results show that the proposed TokenSqueeze method outperforms previous methods by reducing more tokens while mostly preserving the reasoning accuracy.

**Questions:**

Q1: Why DAST and TrainEffi are not compared for AIME25 and LiveCodeBench? Is it because the originally reported numbers do not contain these two datasets? Is it possible to reproduce their results on these new datasets?
Q2: Right now we have a two step process, in each of which a KL-divergence is used. First it's used to get the "good" sequences for training, then it appear again in the DPO objective. Have you thought about potentially combining them in a single objective that can be jointly optimized?
Q3: When you measure the intra-step probability distribution change, there is an approximation using the accumulative token probability distribution. For these tokens, are you using the generated next steps from the original (probably lengthy) solution? If that is the case, this approximation seems to be a bit limited, as what you want is probably a *sequence-level* KL? i.e., $\forall y,  D_{KL}(Q_\theta(y|s_i) || Q_\theta(y|s_i')) < \epsilon$? Of course this would be intractable to approximate, but I'm curious if you have been thinking about doing this on the sequence level as well.

**Ethical Concerns:**

["NO or VERY MINOR ethics concerns only"]

**Final Justification:**

There was a bit of additional confusion during the rebuttal phase but the authors clarified it nicely.

Therefore I maintain my original assessment of this work and think it should be accepted.

**Quality:**

4

**Strengths And Weaknesses:**

**Strengths**:
S1: The proposed method, i.e., using the probability distribution difference of **next reasoning steps** to measure information loss, is quite interesting and intuitive.
S2: The paper is very well-written and easy to follow in general. Specifically, I found the math formulation discussed in 3.2 quite clear and sound at the same time. All the intuitions are well-explained and the presentation of the experiment settings and results is quite clear in delivering the main messages. The discussion of related work is also quite comprehensive, covering both online and offline methods for training the models to generate more concise reasoning traces.
S3: The experiments are also well designed to showcase the effectiveness of the TokenSqueeze method. I especially like using AUC as an evaluation metric, which is perfect to measure the performance with a clear trade-off (i.e., tokens vs. accuracy). I also found the ablation studies presented in section 4.3 quite insightful.

**Weaknesses**:
W1: The evaluations are only conducted on two small models within the same model family (i.e., `DeepSeek-R1-Distill-Qwen-1.5B/7B `), which makes it a bit unclear how much the findings would transfer to reasoning models in general. While I do understand that the training of large reasoning models (e.g., R1) is prohibitively expensive, but I think training on other model series (especially given the new smaller reasoning models came out in the past few months) would significantly help strengthening the claims;
W2: Having a separate section to discussion the limitations and future work would be great;
W3: There are a couple of minor points that I hope the authors can clarify, which I put under the "questions" section.

---

> ### Author Rebuttal · Authors · 2025-07-30
>
> We sincerely thank you for your valuable feedback and positive evaluation of our work. We have carefully addressed each of your suggestions and questions in the responses below.
>
> **Response to Weakness 1:** To further demonstrate the effectiveness of our model, we conducted experiments using Phi4-Reasoning-14B. The results are presented below:
>
> | Model               | **Dataset** | **Accuracy (%)** | **AUC**  | **Avg. Length** |
> | ------------------- | ----------- | ---------------- | -------- | --------------- |
> | Phi4-Reasoning-14B  | AIME24      | 72.1             | 55.1     | 7768            |
> | TokenSqueeze (Phi4) | AIME24      | **73.8**         | **59.6** | **6326**        |
> | Phi4-Reasoning-14B  | MATH500     | 93.8             | 72.0     | 7644            |
> | TokenSqueeze (Phi4) | MATH500     | **96.0**         | **87.5** | **2921**        |
>
> It is important to note that, due to time constraints, we reducing K in intra-step linguistic refinement from 64 to 16. We expect that further hyperparameter tuning would yield even better performance.
>
>
>
> **Response to Weakness 2:** We will include a dedicated section on limitations and future work in the camera-ready version, if the paper is accepted. Some of the limitations and future works we plan to include are listed below:
>
> 	1. Heuristic Hyperparameter Selection: Certain hyperparameters (e.g., the KL threshold $\varepsilon$ used in intra-step rewriting) were selected heuristically based on preliminary experiments. We did not conduct a systematic analysis. In future work, we plan to investigate how varying $\varepsilon$ affects the trade-off between semantic fidelity and brevity.
> 	2. Offline-only Setting: The current pipeline is restricted to offline preference optimisation. We plan to extend it to online reinforcement learning to enable tighter integration between data generation and model updates, potentially improving the trade-off between reasoning efficiency and accuracy.
>
>
>
> **Response to Question 1:** At the time of submission, DAST and TrainEffi had not provided results on the AIME25 and LiveCodeBench datasets in their papers, and neither released training code or model checkpoints. However, in June, DAST made its model publicly available. We evaluated their model on AIME25 and LiveCodeBench, and report the results below:
>
> | Model        | Dataset       | Accuracy (%) | AUC      | Avg. Length |
> | ------------ | ------------- | ------------ | -------- | ----------- |
> | DAST         | AIME25        | 39.1         | 33.6     | **4602**    |
> | TokenSqueeze | AIME25        | **39.8**     | **34.1** | 4711        |
> | DAST         | LiveCodeBench | 29.7         | 26.6     | 3494        |
> | TokenSqueeze | LiveCodeBench | **35.0**     | **31.6** | **3200**    |
>
> Our model outperforms DAST in both accuracy and AUC on AIME25 and LiveCodeBench, with especially notable gains on LiveCodeBench, demonstrating stronger generalization.
>
>
>
> **Response to Question 2:** The two KL terms serve different purposes. In the intra-step rewriting, the KL divergence aims to ensure that the semantic information of the data is preserved before and after rewriting. In contrast, the KL divergence in the training objective is used to constrain the model from deviating too far from the original model (to prevent training collapse). Given the multi-stage training process (data collection → data rewriting → model training), it is currently challenging to combine these two KL terms into a single joint objective. However, for online reinforcement learning methods such as PPO and GRPO, data collection and model training occur simultaneously during the RL process, and we believe there is potential to use a unified optimization objective. We will continue to explore this alignment in future work.
>
>
>
> **Response to Question 3:** In Equation $2$:
> $$
> D_{\mathrm{KL}}^{\text{full}}
> \approx
> \sum_{j=1}^{\min(T,L)}
> D_{\mathrm{KL}}\left(
> Q_{\theta}(\cdot \mid p, s_{\leq i}, t_{1:j-1}) \,\|\, Q_{\theta}(\cdot \mid p, s_{<i}, s_i', t_{1:j-1})
> \right),
> $$
>
> We clarify that $t_j$ refers to the $j$-th token in the consecutive reasoning steps following the $i$-th step $s_i$. Specifically, $t_1$ corresponds to the first token of $s_{i+1}$. **If the tokens in $s_{i+1}$ are insufficient to fill the window, we continue accumulating tokens from subsequent steps until the window is filled or the response ends.**
>
> In fact, KL divergence at the token level can be used as an **unbiased approximation for sequence-level KL divergence**. In Equation 1, we use sequence-level KL divergence, while in Equation 2, we approximate it using token-level KL divergence. Below is a proof that shows how we transition from sequence-level to token-level KL divergence.
>
> **Proof (from Equation (1) to Equation (2))**
>
> ---
>
> **Notation and Setup**
>
> We begin by stating the two key KL divergence expressions from the paper:
>
> - Equation (1):
>   $$
>   D_{\mathrm{KL}}\left( P_{\theta}(\cdot \mid p, s_{\leq i}) \| P_{\theta}(\cdot \mid p, s_{<i}, s_i') \right)
>   $$
>
> - Equation (2):
>   $$
>   D_{\mathrm{KL}}^{\text{full}}
>   \approx
>   \sum_{j=1}^{\min(T,L)}
>   D_{\mathrm{KL}}\left(
>   Q_{\theta}(\cdot \mid p, s_{\leq i}, t_{1:j-1}) \| Q_{\theta}(\cdot \mid p, s_{<i}, s_i', t_{1:j-1})
>   \right)
>   $$
>
> These expressions measure the change in predicted continuation distributions before and after rewriting step $s_i$.
>
> We now define the key variables and distributions involved in the KL computation:
>
> - $p$: the input prompt to the model.
> - $s_i$: the $i$‑th reasoning step generated by the model.
> - $s_{\le i}$: the reasoning trace up to and including step $i$ (i.e., $s_1, s_2, \dots, s_i$).
> - $s'_i$: a rewritten version of step $s_i$ used in intra-step refinement.
> - $t_j$: the $j$‑th token **after** $s_i$ (or $s'_i$), starting the continuation sequence.
> - $t_{1:T}$: a full continuation sequence of $T$ tokens (including EOS).
>
> We define the following probability distributions:
>
> - $Q_\theta(\cdot \mid \cdot)$: the model’s **next-token distribution** under the given context (token-level), parameterized by $\theta$.
> - $P_\theta(\cdot \mid \cdot)$: the model’s **sequence-level distribution** over continuations $t_{1:T}$, given the input context.
>
> To simplify notation, we define:
>
> - $\mathsf{A}(t_{1:T}) := P_\theta(t_{1:T} \mid p, s_{\le i})$
> - $\mathsf{B}(t_{1:T}) := P_\theta(t_{1:T} \mid p, s_{< i}, s'_i)$
>
> These represent the two sequence distributions compared in the KL divergence.
>
> ---
>
> **Step 1: Sequence‑Level KL Definition**
>
> The KL term in Equation (1) is:
> $$
> D\_{\mathrm{KL}}\big( \mathsf{A} \| \mathsf{B} \big)=\mathbb{E}\_{t\_{1:T} \sim \mathsf{A}}
> \left[
> \log \frac{ \mathsf{A}(t\_{1:T}) }{ \mathsf{B}(t\_{1:T}) }
> \right].
> $$
>
> Expanding both $\mathsf{A}$ and $\mathsf{B}$ autoregressively using $Q_\theta$:
> $$
> \begin{aligned}
> \mathsf{A}(t\_{1:T}) = \prod\_{j=1}^{T} Q\_\theta(t\_j \mid p, s\_{\le i}, t\_{1:j-1}), \\
> \mathsf{B}(t\_{1:T}) = \prod\_{j=1}^{T} Q\_\theta(t\_j \mid p, s\_{< i}, s'\_i, t\_{1:j-1}).
> \end{aligned}
> $$
>
> Substituting into the KL definition:
> $$
> \begin{aligned}
> D\_{\mathrm{KL}}(\mathsf{A} \| \mathsf{B})
> &= \mathbb{E}\_{t\_{1:T} \sim \mathsf{A}}
> \left[
> \sum\_{j=1}^{T}
> \log
> \frac{
> Q\_\theta(t\_j \mid p, s\_{\le i}, t\_{1:j-1})
> }{
> Q\_\theta(t\_j \mid p, s\_{< i}, s'\_i, t\_{1:j-1})
> }
> \right] \\
> = \sum\_{j=1}^{T}
> \mathbb{E}\_{t\_{1:j-1} \sim \mathsf{A}}
> \left[
> D\_{\mathrm{KL}}\left(
> Q\_\theta(\cdot \mid p, s\_{\le i}, t\_{1:j-1})
> \Big\|
> Q\_\theta(\cdot \mid p, s\_{< i}, s'\_i, t\_{1:j-1})
> \right)
> \right].
> \end{aligned}
> $$
>
> This shows that the **sequence-level KL** equals the expected sum of per-token conditional KL divergences.
>
> ---
>
> **Step 2: Monte Carlo Approximation**
>
> Computing the full expectation over all possible token histories $t_{1:j-1}$ is intractable in practice. To address this, we use a Monte Carlo approximation: instead of averaging over all continuations, we compute the KL terms along an existing sampled trajectory $t_{1:T}$ drawn from $\mathsf{A}$:
> $$
> D\_{\mathrm{KL}}^{\text{full}}
> \approx
> \sum\_{j=1}^{T}
> D\_{\mathrm{KL}}\left(
> Q\_\theta(\cdot \mid p, s\_{\le i}, t\_{1:j-1})
> \Big\|
> Q\_\theta(\cdot \mid p, s\_{< i}, s'\_i, t\_{1:j-1})
> \right).
> $$
>
> ---
>
> **Step 3: Token-Window Truncation**
>
> To reduce cost further, we truncate the sum at a fixed window size $L$ (e.g., 512 tokens):
> $$
> D\_{\mathrm{KL}}^{\text{full}}
> \approx
> \sum\_{j=1}^{\min(T, L)}
> D\_{\mathrm{KL}}\left(
> Q\_\theta(\cdot \mid p, s\_{\le i}, t\_{1:j-1})
> \Big\|
> Q\_\theta(\cdot \mid p, s\_{< i}, s'\_i, t\_{1:j-1})
> \right).
> $$
>
> This is exactly the formulation given in **Equation (2)** of the main paper.
>
> The approximation introduces a residual term:
> $$
> R\_L = \sum\_{j > L}
> \mathbb{E}\_{t\_{1:j-1} \sim \mathsf{A}} \left[
> D\_{\mathrm{KL}}(\cdot \| \cdot)
> \right] \ge 0,
> $$
> which is small when the effect of rewriting $s_i$ fades over time.
>
> ---
>
> If the paper is accepted, we will include the full proof in the appendix of the camera-ready version.

---

> > ### Comment · Reviewer_JKMo · 2025-08-05
> > **Thanks for the response**
> >
> > I would like to first thank the authors for the new results and detailed discussion.
> >
> > * The new results look great and quite convincing, thanks for getting them in such a short period of time. I think including them in the next version of the paper would make it stronger, so please consider doing so;
> > * Regarding Eq. (1) -> Eq. (2), I guess my question is not really about whether the estimation is biased or not, and it's more about whether you can accurately (i.e., with lower variance) estimate the sequence space KL (Eq.1) with token-level KL. The deduction makes sense to me, except for "Step 2: Monte Carlo Approximation", where if I'm understanding correctly, only a single sampled trajectory is used to estimate this? How is this "Monte Carlo" if only one simulation is done? To lower the variance, should we consider using more sequences and do the average?

---

> > > ### Author Response · Authors · 2025-08-06
> > >
> > > We will include the new results in the next revision of the paper.
> > >
> > > Regarding the concern about the Monte Carlo approximation, we have experimented with larger sample sizes (N > 1). While this does lead to marginal improvements in stability, it also incurs significantly higher computational costs. Given this trade-off, we opted to use N = 1, which also demonstrates strong empirical performance in our experiments.
> > >
> > > In our case, the task involves fine-tuning a reasoning LLM that generates long responses. This setup allows us to assume relatively weak dependencies between tokens within a fixed-length context window. Specifically, although the tokens within a window may appear locally sampled, their generation is in fact heavily influenced by preceding tokens outside the current window. This forms a weakly correlated setting, where the strict independence assumption required by standard Monte Carlo estimation is somewhat relaxed. Despite this theoretical limitation, we find that the approximation remains effective in practice.
> > >
> > > Consequently, the token-level KL divergence is estimated over a local token window (e.g., a 512-token sample, typically an order of magnitude smaller than the full response length), rather than over the full sequence distribution. Although only a single full trajectory is sampled, the KL estimation spans many token positions within that sequence—effectively applying the Monte Carlo approximation across multiple tokens. In summary, this approach represents a practical trade-off between computational cost and estimation efficiency. That said, we agree that using multiple sampled sequences would further reduce the variance of the estimation, albeit at a significant computational expense.

---

### Decision · Program_Chairs · 2025-09-17

**Decision:**

Accept (poster)

**Comment:**

This paper presents TokenSqueeze, a novel method to make reasoning LLMs more efficient by shortening their long chain-of-thought outputs without hurting accuracy. The approach works by first selecting the best reasoning paths and then refining the language in each step to be more concise, using a KL-divergence constraint to ensure the core logic remains intact. The model is then fine-tuned to prefer these shorter, correct answers. The core idea is quite intuitive and the paper is very well-written, with strong experimental results demonstrating significant token reduction. While the initial submission's evaluation was a bit limited to a specific model family and could have benefited from a deeper analysis of some hyperparameters, the authors have addressed these points in their rebuttal with new experiments and clarifications. Overall, this is a solid paper that tackles an important problem in a practical way, and I'm happy to recommend it for acceptance.